# Towards Foundation Models for Mixed Integer Linear Programming

**Sirui Li**[1,*] **Janardhan Kulkarni**[2], **Ishai Menache**[2], **Cathy Wu**[1], **Beibin Li**[2]

[1]MIT, {siruil, cathywu}@mit.edu
[2]Microsoft Research, {jakul, ishai, beibin.li}@microsoft.com

## ABSTRACT

Mixed Integer Linear Programming (MILP) is essential for modeling complex decision-making problems but faces challenges in computational tractability and interpretability. Current deep learning approaches for MILP focus on specific problem classes and do not generalize to unseen classes. To address this shortcoming, we take a foundation model training approach, where we train a single deep learning model on a diverse set of MILP problems to generalize across problem classes. As existing datasets for MILP lack diversity and volume, we introduce *MILP-Evolve*, a novel LLM-based evolutionary framework that can generate a large set of diverse MILP classes with an unlimited number of instances. We study our methodology on three key learning tasks that capture diverse aspects of MILP: (1) integrality gap prediction, (2) learning to branch, and (3) a new task of aligning MILP instances with natural language descriptions. Our empirical results show that models trained on the data generated by *MILP-Evolve* achieve significant improvements on unseen problems, including MIPLIB benchmarks. Our work highlights the potential of moving towards a foundation model approach for MILP that can generalize to a broad range of MILP problem classes. Our code and data are publicly available at `https://github.com/microsoft/OptiGuide`.

## 1 INTRODUCTION

Mixed Integer Linear Programming (MILP) is a versatile mathematical framework widely used to model complex decision-making problems across various domains, including healthcare (Eriskin et al., 2024), supply chain management (Kaya & Urek, 2016), energy systems (Wouters et al., 2015), and finance (Mansini et al., 2015). Its ability to represent intricate combinatorial structures and constraints makes it an indispensable tool in both academic research and industry applications.

Despite its widespread applicability, MILP faces two fundamental limitations. First, tractability is a major concern; solving MILP problems is computationally intensive and time-consuming, particularly for large-scale instances, due to the inherent NP-hardness. This complexity often results in long solve times, posing challenges for time-sensitive applications. Second, understanding the MILP formulation of an optimization problem requires significant expertise in mathematical modeling and optimization, which limits accessibility for non-experts.

To address these challenges, researchers have attempted to leverage machine learning (ML) to enhance MILP solvers, such as learning heuristics to accelerate the solving process (Gasse et al., 2019; Khalil et al., 2016). The underlying hypothesis for why ML may help in MILP is that in most real-world applications, the instances are coming from an unknown distributions, which, although hard to represent analytically, can be learned by deep neural networks. Despite this plausibility, the state-of-the-art ML approaches have achieved limited generalizability due to difficulties in adapting learned models to unseen problems and handling distributional shifts. Most existing methods (Prouvost et al., 2020; Gasse et al., 2022) focus on *specific* problem classes such as Set Cover, Capacitated Facility Location, or Maximum Independent Set, and train class specific models. Unfortunately, these problem specific models fail to perform well on different or more complex MILP instances (Huang et al., 2024). For example, a model trained to help solve Set Cover instances may not generalize

---
*Work done during the author's internship at Microsoft Research

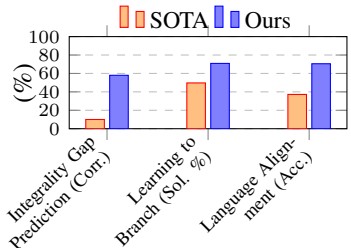
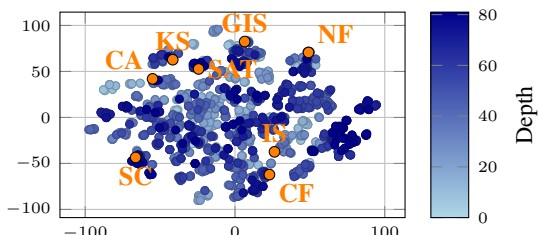

(a) Comparison of performance across tasks for the SOTA baseline models and ours.

(b) Visualization of classes (T-SNE of code embedding): orange points with annotated labels represent seed data, while blue points represent data generated by *MILP-Evolve*, with brightness corresponding to different depths.

Figure 1: Comparison of baseline and our model's performance across tasks, and visualization of *MILP-Evolve* generated MILP classes and seed MILP classes from prior work.

to instances of slightly modified problem formulations (e.g., more constrained versions) as well as other problem classes, such as Facility Location. This lack of generalization hinders practical adoption in real-world applications where problem characteristics vary widely. A concurrent work goes beyond the problem specific training approach (Huang et al., 2024), but the authors consider training a joint model on a small number of selected classes (five), which limits its general applicability.

The state of ML adoption for MILP is in sharp contrast to areas like computer vision and NLP, where the fields have moved away from training problem or task specific models to general purpose foundation models trained on diverse and large-scale datasets that are capable of generalizing to a wide range of tasks. There are several challenges towards building such general purpose foundation models for MILP, the chief among them being the lack of diverse and large-scale training data. Reliance on limited datasets like MIPLIB (Gleixner et al., 2021) is insufficient to capture the vast diversity of MILP problems encountered in practice. More significantly, the distribution of the combinatorial structures of optimal solutions for MILP instances is more complex than the distribution of images or language, challenging the efficacy and development of general purpose foundation models.

Our work is motivated by the following questions:

*Do principles of foundation model training extend to the MILP modality? Can a single model trained on diverse MILP problem classes generalize to unseen MILP classes?*

**The main contributions of this work are highlighted as follows.**

**A Foundation Model Approach for Efficient Multi-Class MILP Learning.** This work is the first to propose a foundation model training approach for MILP learning and demonstrate that, a single model, trained on sufficiently diverse MILP problems, can effectively generalize to a variety of unseen MILP classes. We develop a comprehensive framework that integrates Large Language Models (LLMs) (Achiam et al., 2023; Dubey et al., 2024; Brown et al., 2020) for generating larger and richer training data with Graph Neural Networks (GNNs), which have proven effective in representing MILP instances (Gasse et al., 2019; Scavuzzo et al., 2022; Zhang et al., 2024a). Unlike prior work that trains GNNs on a limited set of MILP classes, we significantly extend the scope by learning a joint model on a broader range of MILP problem classes.

**MILP-Evolve: A Scalable Framework for Generating Diverse MILP Data.** As alluded earlier, unlike text and image modality, existing public MILP datasets lack diversity and volume. To address this limitation, we introduce *MILP-Evolve*, a novel LLM-based data generation pipeline, which leverages frontier models to generate diverse MILP classes and instances from few example seed classes. *MILP-Evolve* combines diverse prompting tailored to the MILP domain, along with parameter search and filtering, to generate a wide range of MILP classes resembling various real-world optimization scenarios. Figure 1b shows that MILP problems generated by our framework have rich diversity. Quantitatively, our approach has enabled us to generate *more than a thousand* different MILP problem classes, much more than any publicly available dataset.

**Comprehensive Framework Evaluation.** We rigorously evaluate our framework across three challenging learning tasks that test different facets of understanding and solving MILP instances in a

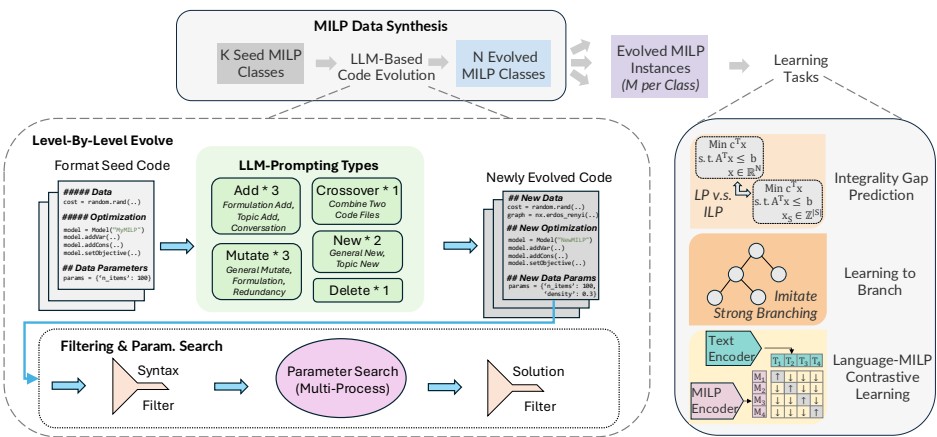

Figure 2: Overview: Left—*MILP-Evolve* pipeline. We design an evolutionary pipeline to leverage a Large Language Model to generate a diverse set of MILP classes. Right—the learning tasks. We design three learning tasks capturing different aspects of MILPs for multi-class learning.

multi-class learning setting. First, we consider the problem of estimating the *integrality gap* (IG) of MILPs, defined as the difference between the optimal value of the MILP and its linear programming relaxation. Our second task, *learning to branch*, is a sequential decision-making task aimed at accelerating MILP solvers by learning efficient branching strategies, a core technique for solving MILPs. While previous works have explored similar tasks (Chen et al., 2023; Geng et al., 2023; Gasse et al., 2019; Zhang et al., 2024a), they primarily focused on training problem-specific models that could only generalize to small set of problem classes. In contrast, we train a single model that can solve each task across a broad range of MILP classes and problems.

We further introduce a new learning task designed to improve the interpretability of MILP instances by mapping each MILP instance in mathematical form to one of several available natural language descriptions. This task tackles a key challenge in MILP accessibility: many open-source MILP datasets and business scenarios provide only raw constraint and variable values, making them difficult to interpret without domain expertise. To address this, we develop a contrastive learning approach that aligns MILP instance embeddings with natural language descriptions, which significantly outperforms direct interpretation attempts by LLMs like GPT-4o. This task complements the tractability tasks by lowering the entry barrier for non-experts and potentially also aiding experts by deepening their understanding of the optimization problems.

**Substantial Multi-Class Learning Gains with Broader Impacts.** Our extensive experiments demonstrate that *MILP-Evolve* combined with our GNN-based learning framework are highly effective in improving multi-class MILP learning. Key results from the held-out test set include a 5.8x correlation improvement for Integrality Gap Prediction, a 1.92x accuracy improvement for MILP alignment, and a 1.42x improvement in the fraction of problems solved to optimum within the time limit for Learning to Branch (see Figure 1a). We further observe improved transfer learning to a separate *MILP-Evolve* test set based on unseen problem classes, as well as on the MIPLIB test set when pre-training on the enriched *MILP-Evolve* dataset. These results highlight the substantial performance gains of our models trained on diverse problem classes generated by *MILP-Evolve*. Our findings reveal key insights for advancing MILP learning: *the diversity of training data has substantially greater impact on model performance than data quantity alone.* These insights, along with our study's broader contributions, represent significant progress towards foundation models for MILPs.

## 2 FORMAL DESCRIPTIONS OF MILP AND LEARNING TASKS

### 2.1 MIXED INTERGER LINEAR PROGRAMMING (MILPs)

MILP involves optimization problems where some decision variables are constrained to be integers, and relationships among variables are linear in the form of constraints and an objective function. A

MILP problem is formulated as:

$$x_{ILP}^* = \arg\min\{c^\intercal x : Ax \le b, x_j \in \mathbb{Z} \forall j \in I\}, \tag{1}$$

where $x \in \mathbb{R}^n$ is the vector of decision variables, $c \in \mathbb{R}^n$ is the cost vector, $A \in \mathbb{R}^{m \times n}$ and $b \in \mathbb{R}^m$ define the linear constraints, and $I \subseteq \{1, \dots, n\}$ indicates the indices of variables required to be integer-valued. MILP is extensively used to model complex decision-making problems involving both discrete choices and continuous variables, capturing applications in supply chain optimization, scheduling, network design, and resource allocation (Duong & Bui, 2018; Floudas & Lin, 2005; Rieck et al., 2012). However, solving MILP problems is challenging due to their NP-hardness, leading to computational intractability for large-scale instances.

## 2.2 LEARNING TASKS

We give formal definitions of the three tasks considered in this work, each captures a different but interconnected aspect of understanding and solving MILPs.

### 2.2.1 INTEGRALITY GAP PREDICTION

Integrality gap (IG) quantifies how close the linear programming (LP) relaxation of a MILP instance is to its integer optimum. A smaller gap indicates that the LP relaxation is a good approximation, while a larger gap suggests a more challenging problem requiring more computational resources. When the integrality gap is small, one can use the optimal solution to the LP relaxation and round it to obtain integral solutions, which is a foundational technique in the design of approximation algorithms (Raghavan & Tompson, 1987; Williamson & Shmoys, 2011).

For each MILP instance $x$, we compute the integrality gap as: $g^*(x) = \frac{|z_{ILP}^*(x) - z_{LP}^0(x)|}{|z_{ILP}^*(x)|}$, where $z_{LP}^0(x)$ is the LP relaxation value at the root node, and $z_{ILP}^*(x)$ is the optimal solution value of the MILP (see Appendix A.2.1 for details). Accurately predicting $g^*(x)$ allows us to estimate the difficulty of solving the MILP before actually solving it. When the integrality gap is small, a good approximation of $g^*(x)$ can quickly provide a good estimate for the MILP's optimal value, as LPs can be efficiently solved in practice; it further suggests that MILP solvers may converge more quickly to optimal solutions, making it a potential indicator of faster solve times for the corresponding MILPs.

### 2.2.2 LEARNING TO BRANCH

Branching decisions in the Branch-and-Bound (B&B) algorithm (Land & Doig, 2010) significantly impact the efficiency of solving MILPs. Selecting the right variable to branch on can reduce the size of the search tree and, consequently, the computation time. Traditional strategies like strong branching (Applegate et al., 1995; Achterberg, 2007) are effective but computationally expensive.

By learning a branching policy through deep networks, the goal is to approximate the performance of strong branching at a small fraction of the computational cost. Specifically, at each B&B node $(s, \mathcal{A}(s))$, where $s$ is the MILP representation at the node and $\mathcal{A}(s)$ is the set of candidate variables to branch on, the learning task is to imitate the strong branching expert, which selects the action $a^* \in \mathcal{A}(s)$. We minimize the cross-entropy loss $-\frac{1}{N} \sum_{(s,a^*)} \log \hat{f}_\theta(a^*|s)$ of predicting the expert action, given the network $\hat{f}_\theta(\cdot)$ (see Appendix A.2.2 for details). Building on the work of (Gasse et al., 2019), this sequential decision-making task aims to enhance solver efficiency, making it more practical for large-scale or time-sensitive applications.

### 2.2.3 LANGUAGE-MILP CONTRASTIVE LEARNING

Understanding and interpreting MILP instances, given by the numerical $A, b, c$ values, is inherently challenging due to their abstract and mathematical nature. To make MILP problems more accessible and to facilitate interaction with optimization problems as a new modality, we propose a contrastive learning task that aligns MILP instances with their corresponding natural language descriptions.

We design and employ a contrastive learning framework inspired by the Contrastive Language-Image Pretraining (CLIP) framework (Radford et al., 2021). Specifically, we minimize a symmetric cross-entropy loss over our dataset $\mathcal{D}$, which encourages the model to associate each MILP instance with its corresponding text description and vice versa.

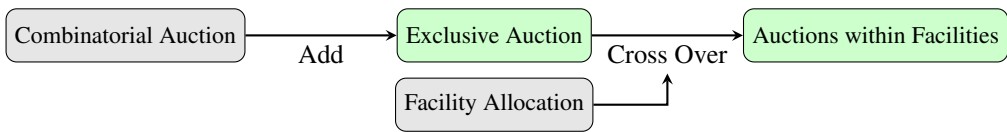

Figure 3: An evolution chain for an auction problem consisting of two iterations. In the first iteration of the evolution, we add a constraint. The second iteration involves a crossover with the facility allocation problem. The details of code generated by LLM is in Appendix Figure 6.

Formally, let $\mathcal{D} = \{(M_i, T_i)\}_{i=1}^N$ denote a dataset comprising $N$ pairs of MILP instances $M_i$ and their corresponding textual descriptions $T_i$. Our objective is to learn embedding functions $f_M : \mathcal{M} \to \mathbb{R}^d$ and $f_T : \mathcal{T} \to \mathbb{R}^d$ that map MILP instances and text descriptions into a shared $d$-dimensional latent space. The goal is to ensure that the embeddings of matching pairs $(M_i, T_i)$ are closer to each other than those of non-matching pairs, effectively capturing the semantic relationships between MILPs and their descriptions. Details are provided in Appendix A.2.3.

By optimizing the contrastive loss, the model learns to project MILP instances and their textual descriptions into a common embedding space where semantically related pairs are close together. This learning task can serve an important stepping-stone towards the more challenging task of directly generating natural language descriptions from MILP instances. This is in similar spirit to how the CLIP framework (Radford et al., 2021) paved the way for text to image generative models such as DALL·E (Ramesh et al., 2021) by training a decoder model on CLIP embeddings.

**Enhancing MILP applicability.** The three learning tasks are complementary and collectively essential for enhancing the applicability of MILP through *understanding, predicting, and accelerating*. In particular, the Language-MILP task helps understand the structure and properties of MILP instances, aiding non-experts in problem comprehension and may further assist experts by deepening their understanding of the problems; the Integrality Gap Prediction task focuses on analyzing solution properties of the MILP instance, potentially allowing instances with tight gaps to be solved via LP relaxation, coupled with rounding algorithms, without fully solving the MILP; the learning to Branch task enhances MILP solving efficiency through more effective branching, which can have huge time and cost savings in industrial applications.

## 3 MILP-EVOLVE FOR GENERATING DIVERSE MILP CLASSES

The diversity of MILP problems is a crucial aspect for using a foundation model approach for MILP. Existing studies, however, often focus on a small number of problem classes, limiting the model's ability to generalize. We propose *MILP-Evolve*, an LLM-based pipeline designed to generate a diverse set of MILP classes. By starting from a small set of seed classes and systematically generating new ones, we aim to capture a broader spectrum of optimization problems.

**Class representation.** We represent each class of MILP problem with a modularized optimization code script, comprising three primary functions: (1) **Data**, responsible for generating the necessary data to construct the objective and constraint coefficients, often from uniform distributions or common graph structures; (2) **Optimization Modeling**, which utilizes this data to define the decision variables, constraints, and the objective function of the optimization problem; and (3) **Parameters**, which outlines the problem-specific parameters for the data distribution, such as the number of nodes or graph density in graph-related problems.

**Seed classes.** We curate a set of eight MILP classes commonly used in prior literature (see Appendix A.1.1) and use them as the starting point to generate a more diverse set of MILP classes.

***MILP-Evolve* – LLM-generated classes.** Generating diverse MILP classes is non-trivial due to potential pitfalls such as generating incorrect code, infeasible instances, or lack of diversity. To overcome these challenges, our pipeline, as illustrated in Fig. 2 (left), consists of two key steps.

*1. Class Generation.* Given the seed class representations, we use LLMs to generate new MILP classes by applying several transformations. These transformations are represented by 10 chain-of-thought (Wei et al., 2022) prompt operators, which are described in detail in Appendix A.1.2. At each evolve iteration, *MILP-Evolve* selects a random subset of these operators and invokes them se-

quentially, where each prompt operator is applied to the previously evolved MILP class. We add each MILP class generated by this process to our *MILP-Evolve* dataset. An example of a two-iteration evolution is shown in Figure 3. Intuitively, MILP classes that are generated in early iterations of the framework are closer to the seed class, whereas those towards the end are further away.

At a high level, the prompt operators include *adding* or *deleting* constraints, variables, or the associated data; *crossing over* classes; *mutating* constraints and/or objectives; and creating entirely *new* classes. Each prompt consists of three main modules that the LLM should follow step-by-step: (1) summarize the given MILP class, (2) describe how to modify the MILP class based on the specific operator type, (3) generate the new MILP class that follows the class representation.

To encourage diversity and realism of the generated classes, we guide the LLM to integrate MILP problems into real-world contexts and generate classes tailored to specific industry needs. This requirement is achieved by either prompting the LLM to base the MILP on real-world applications or directly providing realistic topics, such as "pharmaceutical company planning cross-country vaccine distribution with storage, transport, and demand constraints," when generating the MILP class.

We further prompt the LLM to apply commonly used and advanced MILP formulation techniques, such as *Big M* (Wolsey, 2020), *Special Order Sets* (Beale & Forrest, 1976), and *Symmetry Breaking* (Margot, 2009), to make the generated instances technically challenging. By doing so, we guide the LLMs to produce a wide range of MILP formulations reflecting real-world scenarios.

*2. Filtering.* After generating new MILP classes, we perform filtering and parameter adjustment to ensure the usefulness and mathematical feasibility of the generated instances. Our modular class representation allows us to easily identify and adjust key parameters to achieve desired properties such as feasibility, appropriate solve times, and reasonable problem sizes (see Appendix A.1.4).

**Outcome.** Using our pipeline, with GPT-4o as the LLM, we create a dataset of MILP classes significantly more diverse than those used in prior work; see Figure 1b. Examples of the generated classes and implementation details of the *MILP-Evolve* pipeline can be found in Appendix A.1.

# 4   TRAINING PIPELINE

Our training pipeline (Fig. 2), consists of two main components: a) The *MILP-Evolve* system described above to obtain diverse dataset; b) The learning architecture, which combines GCNs with an attention module, to effectively learn in a multi-class setting. We provide more details below.

*Input.* We represent each MILP instance as a bipartite graph connecting variable nodes $V$ and constraint nodes $C$ (Gasse et al., 2019), capturing the structural relationships within the MILP. This graph serves as input to our neural network architecture. For Language-MILP Contrastive Learning, the text description inputs are embedded via NV-Embed-v1 (Lee et al., 2024), an open-source text embedding model based on Mistral 7B (Jiang et al., 2023).

*Architecture.* For Integrality Gap Prediction and the MILP encoder for Language-MILP Contrastive Learning, our architecture consists of the following main components: 1) We use a Graph Convolutional Network (GCN) (Kipf & Welling, 2017; Gasse et al., 2019; Paulus et al., 2022; Scavuzzo et al., 2024) to embed the variable and constraint nodes, allowing information to flow between them and capturing local structural patterns. 2) To capture global dependencies while reducing computational overhead, we use the attention mechanism (Vaswani, 2017) over a sub-sampled set of variable and constraint nodes. 3) We include three additional nodes for the attention: two representing the aggregated variable and constraint node embeddings (obtained via mean pooling) and a *summary* node used to extract a global representation of the MILP instance. 4) The attention module outputs an updated embedding of the summary node, which is then passed through a final linear layer to produce the final output. Empirically we find that incorporating the attention layers improves performance, especially for transfer learning to the MIPLIB dataset (Sec. 5.3). This result is expected as attention mechanisms allow for better global understanding of MILP instances.

For the learning to branch task, we directly use the variable embeddings produced by the GCN to predict the variables selected for branching, building upon a similar architecture employed in Gasse et al. (2019). As the model needs to be invoked for each Branch-and-Bound node, we omit the attention layer to reduce the computational overhead.

Table 1: Out-Domain Performance on the *MILP-Evolve* held-out test set. *Ours* outperforms the baseline methods in all three learning tasks.

(a) Integrality Gap Prediction.

|  | Deviation ($\downarrow$) | Corr. ($\uparrow$) |
|---|---|---|
| **Mean** | 33.07% | 0.00 |
| **Seed** | 32.96% | 0.10 |
| **Seed + Param.** | 32.77% | 0.07 |
| **Seed + VAE** | 30.27% | 0.26 |
| **Ours - Attn.** | 20.82% | 0.57 |
| **Ours** | **20.14%** | **0.58** |

(b) Language-MILP Contrastive Learning.

|  | 4-Way Acc. ($\uparrow$) | 10-Way Acc. ($\uparrow$) |
|---|---|---|
| **Seed** | 37.21% | 18.73% |
| **Seed + Param.** | 39.68% | 20.35% |
| **Seed + VAE** | 33.67% | 15.58% |
| **Ours - Attn.** | 70.41% | 52.76% |
| **Ours** | **70.54%** | **54.17%** |

(c) Learning to Branch. For each method, we report the proportion of instances solved to optimal within a $200s$ time limit (including network overhead); among those instances, we report the 1-shifted geometric mean of time improvement over SCIP Default including and excluding (Ex.) the network overhead (See Appendix A.2.2).

|  | Time Improv. ($\uparrow$) | Time Improv. (Ex) ($\uparrow$) | % Instances Solved to Optimal ($\uparrow$) |
|---|---|---|---|
| **Seed (GCN)** | -30.65% | -16.60% | 49.59% |
| **Seed + Param. (GCN)** | -3.70% | 5.89% | 64.52% |
| **Seed + VAE (GCN)** | -26.18% | -13.50% | 49.06% |
| **Ours (GCN)** | **15.49%** | **21.02%** | **70.90%** |

## 5 EXPERIMENTS

To evaluate the effectiveness and generalization capabilities of our approach, we conduct comprehensive experiments focusing on out-of-domain settings where models are tested on MILP classes unseen during training. For each learning task, we train a single model on the diverse MILP classes generated by *MILP-Evolve*. The evaluation is performed on two fronts: first, on a held-out set of MILP classes from *MILP-Evolve*, and second, through transfer learning on instances from MIPLIB, a widely recognized MILP benchmark dataset.

### 5.1 EXPERIMENTAL SETUP

More than a thousand MILP problem classes are used, with the exact number varying by task. For IG prediction, we excluded instances where the optimal solution was not found within a time limit of 200s. For the learning to branch task, our training data consists of collecting strong branching expert examples that are obtained by solving MILP instances up to the same time limit of 200s (see details in Appendix A.1.5).

**State-of-the-art Baselines.** To assess the effectiveness of training on the diverse MILP classes generated by *MILP-Evolve*, we compare our model against several methods. We collected a **Seed** dataset containing all problem classes used in recent state-of-the-art (SOTA) studies (Gasse et al., 2019; Scavuzzo et al., 2022; Labassi et al., 2022), totaling 16 sets with two parameters for each of the eight existing classes. We also create a competitive baseline, **Seed + Param.**, which expands the Seed dataset by including 89 additional parameters identified through a parameter search procedure similar to that used in *MILP-Evolve*. The third approach, **Seed + VAE** (Guo et al., 2024), employs a Variational Autoencoder (VAE)-based instance generation method to augment the seed class instances, which further incorporates constraint grouping to improve upon the seminal work of (Geng et al., 2023). Details of these baselines can also be found in Appendix A.1.5. This comparison emphasizes the distinction between our MILP class augmentation approach and existing instance augmentation methods. For integrality gap prediction, we also include a **Mean** baseline, which, for all MILP instances, predicts the same constant value given by the mean of all the training set labels. For Language-MILP Alignment, we include a further comparison with GPT-4o in Appendix A.3.7.

For simplicity, we refer to the model trained with data generated from *MILP-Evolve* as *Ours*. Additionally, to evaluate the impact of our architectural choices, we include a comparison with our model trained without the attention layer, referred to as *Ours - Attn*. Details on architecture, training, metrics, and additional ablation results are provided in Appendices A.2 and A.3.

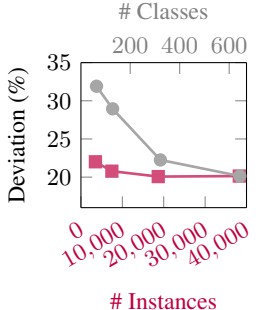 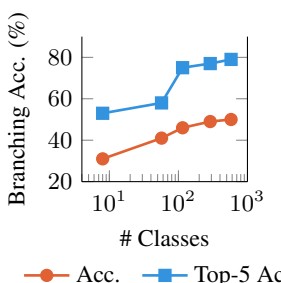 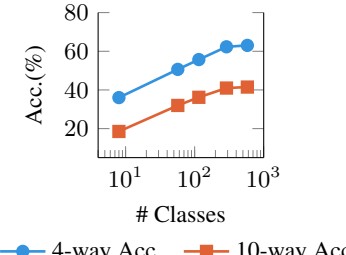

| | | |
|---|---|---|
| (a) Integrality Gap Prediction. Purple: Different number of instances per class, same number of classes. Gray: Different number of classes, same number of instances per class. | (b) Learning to Branch. We keep the total number of training instances constant and vary the number of training MILP classes. We plot the prediction accuracy with respect to the strong branching expert. | (c) Language-MILP Contrastive learning. We control the total number of training instances to be the same (1000 instances) across experiments and change the number of training MILP classes. |

Figure 4: Effect of Scaling Training Classes on Test Performance: Increasing training classes improves test performance across all learning tasks.

Table 2: Transfer Learning Results on a *MILP-Evolve* Test Set on Six Unseen Seed Classes. Initializing with the *Ours* pre-trained model yields the best performance (see Appendix A.3.3 for details).

| | Integrality Gap | | Learning to Branch | | Language-MILP | |
|---|---|---|---|---|---|---|
| | Deviation ($\downarrow$) | Correlation ($\uparrow$) | Acc. ($\uparrow$) | Top 5 Acc. ($\uparrow$) | 4 Way Acc. ($\uparrow$) | 10 Way Acc. ($\uparrow$) |
| **Train From Scratch** | 21.41% | 0.65 | 28.93% | 69.70% | 72.37% | 46.50% |
| **Seed** | 21.25% | 0.65 | 23.11% | 56.82% | 72.20% | 42.45% |
| **Seed + Param.** | 25.61% | 0.52 | 27.87% | 68.32% | 75.17% | 42.66% |
| **Seed + VAE** | 23.40% | 0.58 | 25.25% | 60.81% | 72.90% | 44.61% |
| **Ours** | **17.98%** | **0.68** | **30.71%** | **70.33%** | **77.62%** | **53.99%** |

**Metrics.** For the integrality gap task, we compute the mean absolute error (deviation) and Pearson correlation coefficient. To evaluate the learning-to-branch approach, we measure the proportion of instances optimally solved within the time limit, as well as the percentage improvement in solve time compared to the default SCIP solver. The time improvement is presented both with and without accounting for the GPU time required by the deep learning model. In assessing the contrastive learning task, we report both 4-way and 10-way accuracy, where the model must identify the correct text description corresponding to a given MILP instance from a set of 4 or 10 options, respectively.

## 5.2 *MILP-Evolve* TEST SETS

We partition the set of MILP problem classes generated by *MILP-Evolve* into roughly a 7:1:2 split for training, validation, and test subsets which we denote by $\mathcal{X}_{\text{train}}^{\text{Evolve}}$, $\mathcal{X}_{\text{val}}^{\text{Evolve}}$, and $\mathcal{X}_{\text{test}}^{\text{Evolve}}$. For each learning task, we train our model on instances from $\mathcal{X}_{\text{train}}^{\text{Evolve}}$, referred to as *Ours*, and evaluate its performance on the test instances $\mathcal{X}_{\text{test}}^{\text{Evolve}}$. This setup helps us to assess the impact of class diversity and instance variety on the model performance.

Table 1 showcases the performance on the *MILP-Evolve* held-out test set across the three learning tasks. The *Ours* approach consistently outperforms all baselines trained on fewer classes, highlighting the critical role of class diversity in enhancing model generalization. Furthermore, the inclusion of the attention mechanism significantly boosts performance, underscoring its importance in capturing global information within MILP instances. The (Seed + Param) baseline outperformed the learning-based data augmentation method (Seed + VAE) in contrastive learning and branching, suggesting that the VAE approach might not be readily extensible. Moreover, the learning to branch task remains particularly challenging in a multi-class setting, suggesting an area for future improvement and research (see Appendix A.3.2 for details).

Table 3: Transfer Learning Results on MIPLIB. Initializing with the *Ours* pre-trained model yields the best performance on the MIPLIB test set.

| | Integrality Gap Prediction | | Language-MILP Contrastive Learning | |
|---|---|---|---|---|
| | Deviation (↓) | Correlation (↑) | 4 Way Acc. (↑) | 10 Way Acc. (↑) |
| **Train From Scratch** | 28.21% | 0.43 | 73.28% | 65.29% |
| **Seed** | 26.97% | 0.44 | 79.69% | 72.71% |
| **Seed + Param.** | 23.30% | 0.54 | 76.41% | 71.74% |
| **Seed + VAE** | 24.76% | 0.50 | 79.92% | 70.99% |
| **Ours** | **21.56%** | **0.59** | **82.08%** | **75.57%** |

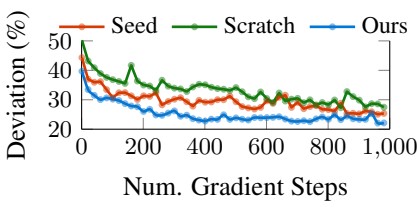

(a) Integrality Gap Prediction.

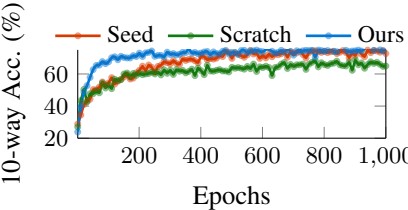

(b) Language-MILP Contrastive Learning

Figure 5: Convergence Rate on the MIPLIB Dataset. We plot the test performance for models at different stages of the training process for transfer learning to MIPLIB. Initializing with *Ours* pre-trained model leads to significantly faster convergence than other baselines.

We also find that *scaling up the number of training classes is much more important than increasing the number of training instances*, emphasizing the importance of diverse MILP classes for training. As illustrated in Figure 4a, reducing the number of classes while maintaining the same total number of instances significantly affects performance, whereas reducing the number of instances per class has a minimal effect. Similarly, we observe in Figures 4b and 4c an upward trend in performance as the number of classes increases, while keeping the total number of training instances constant. These results reinforce the value of *MILP-Evolve* in generating a diverse set of MILP classes.

To further ensure that our held-out test set is representative and can generalize to unseen datasets, We collect another test set of 50 MILP classes, by running *MILP-Evolve* on six *unseen* Seed classes *different from those used in Table 1*. We split the dataset into $\mathcal{X}_{train}^{new}$ and $\mathcal{X}_{Test}^{new}$, fine-tune the pretrained models from Table 1 on $\mathcal{X}_{train}^{new}$ and evaluate the performance on $\mathcal{X}_{test}^{new}$. In Table 2, we see that initializing with *Ours* enables the best transfer learning performance on this new test set.

## 5.3 MIPLIB TRANSFER LEARNING

Finally, we evaluate the transfer learning performance of our model on MIPLIB (Gleixner et al., 2021), a commonly used heterogeneous benchmark dataset *which is never used in our pretraining process*. We filter MIPLIB to include instances with known optimal solution (for gap prediction) or with meaningful description (for contrastive learning), and split them into training and test sets, $\mathcal{X}_{train}^{MIPLIB}$ and $\mathcal{X}_{test}^{MIPLIB}$. We fine-tune our pretrained model from $\mathcal{X}_{train}^{Evolve}$ on $\mathcal{X}_{train}^{MIPLIB}$ and subsequently evaluate its performance on $\mathcal{X}_{test}^{MIPLIB}$ (see Appendix A.1.6 for details).

Table 3 presents the results of transfer learning on the MIPLIB dataset for the IG and CL tasks. First, we observe that pretraining models on a diverse dataset is beneficial compared to training from scratch (first row in the table). Further, our model trained on data from *MILP-Evolve* achieves superior performance compared to all baselines. Instance augmentation methods like VAE and parameter search provide some incremental gains, but are outperformed by our approach. Figure 5 further illustrates the convergence behavior during fine-tuning on MIPLIB. Models pretrained with *MILP-Evolve* not only converge faster, but also achieve better final performance after fine-tuning.

We hypothesize that the enhanced performance during fine-tuning stems from the broad diversity of MILP classes generated by *MILP-Evolve*, which captures a wide range of problem distributions, constraint types, and variable interactions (see Fig. 11 in Appendix A.3.1 for details). This extensive

coverage enables the model to adapt more effectively to the varied structures present in MIPLIB, compared to models trained on more homogeneous or limited datasets.

We omit learning to branch task on MIPLIB dataset because we find that a vast majority of MIPLIB instances take either too little or too much time to solve; in fact, only 13 instances can be solved to optimality with a solve time between $20s$ and $300s$, making the evaluation set too small. We believe that more comprehensive pre-training with more compute are critical for learning to branch in datasets similar to MIPLIB; this is an interesting direction for future work.

## 6 RELATED WORK

**Machine Learning for MILP.** There has been a surge of research using machine learning techniques to improve MILP solvers. Studies cover various aspects of MILP solving, including presolving (Liu et al., 2024a), variable selection for branching (Zhang et al., 2024a; Huang et al., 2024; Scavuzzo et al., 2022; Gupta et al., 2020), node selection for exploration (Labassi et al., 2022; Song et al., 2018; He et al., 2014), generating cutting planes (Wang et al., 2023; Paulus et al., 2022; Tang et al., 2020), configuring solver parameters (Li et al., 2023b; Balcan et al., 2021; Xu et al., 2011), and scheduling primal heuristics (Chmiela et al., 2021; Hendel et al., 2019; Khalil et al., 2017).

Despite these advancements, a significant challenge remains in the *generalizability* of learned models across different tasks and problem classes. Most existing ML methods for MILP focus on specific problem classes (Prouvost et al., 2020; Gasse et al., 2022) and struggle to generalize to unseen types of problems. This limitation hinders practical adoption, as real-world applications often involve a wide variety of MILP formulations.

**Large Language Models for MILP.** Recent research has explored prompting LLMs for MILP and combinatorial optimization tasks, such as optimization modeling (AhmadiTeshnizi et al., 2024), what-if analysis (Li et al., 2023a), infeasibility diagnosis (Chen et al., 2024), and automatic heuristic design (Liu et al., 2024b; Romera-Paredes et al., 2024; Yang et al., 2024). The majority of these works are data-agnostic, usually taking code or human language as input and performing analytical tasks without accessing the critical $A, b, c$ matrices. While many studies have aligned images, voices, code, and genomics modalities for LLMs (Liu et al., 2024c; Barrault et al., 2023; Roziere et al., 2023; Abdine et al., 2024), few have integrated the MILP modality with text.

**MILP Datasets.** For common benchmark MILP classes (Prouvost et al., 2020), the objective and constraint coefficients are typically generated randomly within specified ranges, often representing weights or budgets of variables. For graph-based MILPs (e.g., Set Cover), the constraints and variables $A, b, c$ depend on the underlying graph structure and are generated from distributions like Erdős–Rényi (Erdös & Rényi, 1959) or Barabási–Albert (Albert & Barabási, 2002).

Recent studies use heuristics (Drugan, 2013; Bowly, 2019), Variational Autoencoders (Guo et al., 2024; Geng et al., 2023), or Diffusion Models (Zhang et al., 2024b) to generate MILP instances from a given set. These works focus on *instance-based* MILP generation: given a limited number of instances within a specific MILP class (e.g., Set Cover), they generate *similar* instances to augment the dataset, rather than more *diverse* ones. In contrast, we aim to generate diverse problem classes, including more complex or entirely different problems not found in existing public datasets.

## 7 CONCLUSION

This paper takes a first step towards a foundation model training approach for MILP. We address three key learning tasks and train a single model for each task to generalize across MILP problem classes. We introduce *MILP-Evolve*, a general MILP class generation method that leverages LLMs to enable larger, richer, and more diverse datasets for training. Our experiments demonstrate that our framework, trained with *MILP-Evolve*-generated data, significantly outperforms previous works.

While we advanced the state-of-the-art by using a single model across MILP classes instead of multiple class-specific models, we acknowledge that this work still trains separate models for each learning task. An important future direction is to unify these tasks into a single "foundation" model which would potentially extend to more tasks, such as generating cutting planes, learning solver parameters, and formulating reliable optimization problems from text descriptions.

## 8 ACKNOWLEDGEMENT

The authors would like to thank Luke Marshall, Konstantina Mellou, Marco Molinaro, and Yining Ma for insightful discussions. The authors further thank the anonymous reviewers for their valuable feedback.

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

# A APPENDIX

CONTENTS

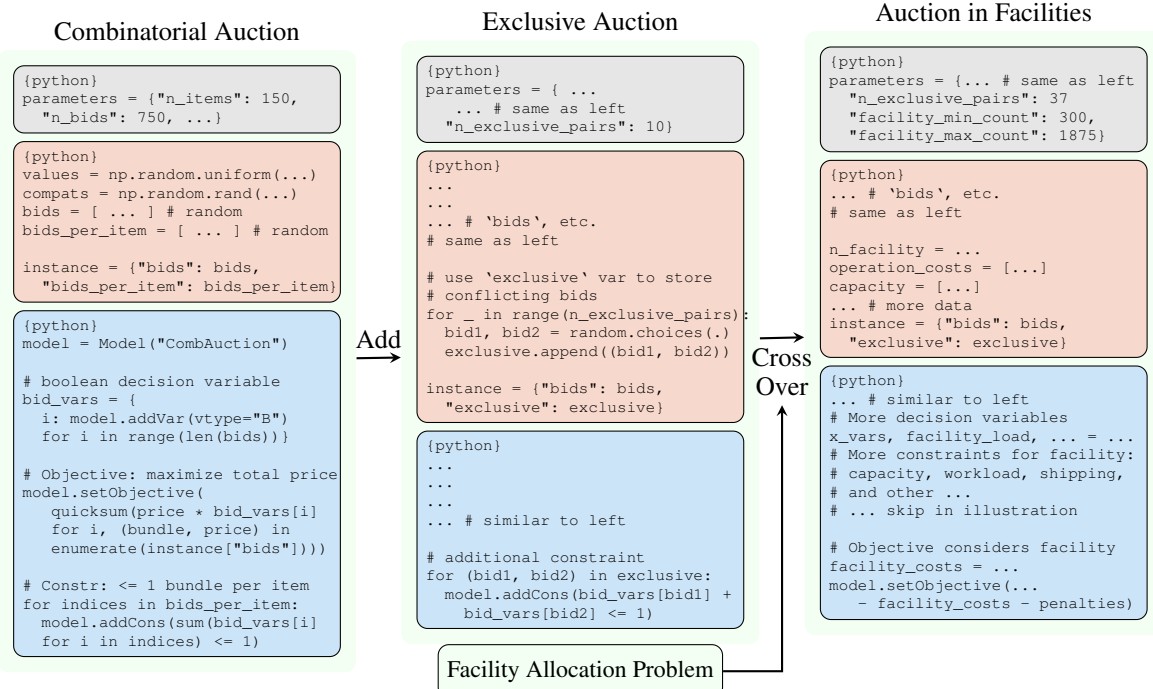

Figure 6: An evolution chain for a combinatorial auction problem. In the first evolution, an additional "mutually exclusive" constraint is introduced for different bids. The second evolution involves a crossover with a facility planning problem, where bids are placed at different facilities, each with its own auction capacity. For each problem class, the components are highlighted in different colors: parameters (gray), data (orange), and solver (blue). For brevity, most of the code is omitted.

## A.1 GPT-BASED EVOLUTION

*MILP-Evolve*, as depicted in Fig. 2, generates MILP code (classes) *level by level*, combining LLM-based code generation with a post-generation parameter search and filtering at each level. The process begins with 8 seed classes taken from previous literature. The code of each seed class is reformatted into a standard, modular code structure including data, optimization modeling, and parameters, as detailed in Appendix A.1.1.

At each level, up to $K = 108$ pairs of (MILP code, prompt type) are sampled to create a new batch of prompts for the LLM. For each pair, the MILP code is sampled uniformly at random from the successfully generated MILP code from the previous level (or seed classes at the first level). A prompt type is then randomly sampled to prompt the LLM, which generates a new MILP code. The prompt type is selected with weights within a set of 10 prompt subcategories across five broad categories (**Add**, **Cross-Over**, **Mutate**, **New**, **Delete**), with further details provided in Appendix A.1.2. We use OpenAI GPT-4o Achiam et al. (2023) as the LLM for the MILP class generation.

Since LLM-generated code can often be buggy, infeasible, or trivially solved within a short time, we implement an (automated) post-generation parameter search and filtering procedure. This includes: removing all buggy generated MILP code, (2) identifying parameters for each remaining MILP code, and (3) conducting a parameter grid search by solving the MILP with various parameter values. A MILP code is considered "successful" if at least one parameter combination meets a set of intuitive criteria for problem size, solve time, and branch-and-bound tree size; otherwise, the code is discarded. Details on the parameter grid-search ranges and filtering criteria are provided in Appendix A.1.4.

**Computation Requirements.** We conduct 92 levels of MILP class evolution, with a total of $9,044$ prompts submitted to GPT-4o. This results in $2,006$ successfully generated MILP classes that meet the filtering criteria. Each evolution level takes approximately 3 hours on average, with a total of

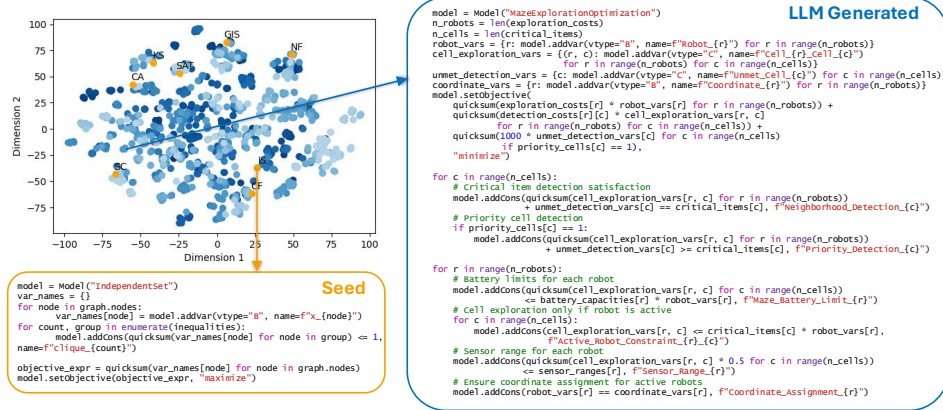

Figure 7: Top Left: The same T-SNE visualization as in Fig. 1a. Bottom Left: an example of a seed class' optimization modeling component. Right: an example of a *MILP-Evolve* generated class' optimization modeling component.

10 days with 60 parallel running threads to generate all levels; the majority of the time is spent on parameter search and filtering, as it involves solving a large number of MILP instances. For the learning experiments, we use the first $1,000$ generated MILP classes, which were produced in the first $42$ levels over approximately $5$ days. We plan to release all LLM-generated classes upon paper acceptance, and we believe this dataset will be a valuable resource for future multi-class MILP learning tasks.

### A.1.1  MILP CLASSES DETAILS

**Seed Classes.**  The evolution pipeline starts with 8 benchmark MILP classes commonly used in previous literature. Table 4 provides a list of abbreviations and references of the seed classes. We reformat the code from (Scavuzzo et al., 2022) to generate IS, SC, CA, CF, and KS instances, and from (Labassi et al., 2022) to generate GIS, NF and SAT instances.

Table 4: Abbreviations, Full Names, and References for the 8 Seed Classes.

| Abbreviation | Full Name | Reference |
|---|---|---|
| IS | Maximum Independent Set | Bergman et al. (2016) |
| SC | Set Cover | Balas & Ho (1980) |
| CA | Combinatorial Auction | Leyton-Brown et al. (2000) |
| CF | Capacitated Facility Location | Cornuéjols et al. (1991) |
| KS | Multiple Knapsack | Pisinger (1999) |
| GIS | Generalized Independent Set | Colombi et al. (2017) |
| NF | Multicommodity Network Flow | Hewitt et al. (2010) |
| SAT | Max Satisfiability | Béjar et al. (2009) |

**Code Syntax.**  We formulate each seed class into a modular code structure, and we also prompt each LLM-generated class to follow a similar structure. A detailed example of an LLM-generated class (`ConferenceRoomScheduling`) provided in Appendix A.4.1. Each MILP class is implemented as a Python class with a descriptive name, with three main components:

1. **Data:** Inside the `generate_instance` function, necessary data for the constraint and objective coefficients are generated. In the conference room scheduling example, the data corresponds to the meeting schedules, room availability and capacities. For many graph based problems (e.g. Set Cover, Independent Set), the data includes a graph generated by common distributions such

as Erdos-Renyi or Barabasi-Albert. The data generation typically uses random functions with parameters to control data properties (e.g. the range of room capacity or the graph density).

2. **Optimization Modeling:** Inside the `solve` function, variables, constraints and the objective function of the MILP is defined. For example, the conference room scheduling example maximizes the number of meetings scheduled, subject to the constraints that each room can host only one meeting at a given time and each meeting should be scheduled in at most one room. The data generated by the `generate_instance` function is used to generate these constraints.

3. **Parameters:** The `parameters = {...}` dictionary specifies parameters for data generation to complete the code for the MILP class.

The code for each MILP class is self-contained and complete; When executed, it can generate data, model the optimization, and solve the corresponding MILP problem.

When prompting the LLM to generate new MILP code, we pre-process each file by marking the data, optimization, and parameters blocks with comments including ### given instance data code ends here, ### new instance data code ends here *(data)* ### given constraints and variables and objective code ends here, ### new constraints and variables and objective code ends here *(optimization modeling)*, and ### given parameter code ends here, ### new parameter code ends here *(parameters)* at the end of each function.

### A.1.2 PROMPT TYPES AND HIGH LEVEL DESCRIPTIONS

**Level-by-Level Prompt Generation** At each level, we randomly sample at most $K = 108$ pairs of (MILP code, prompt type) to form the new batch of prompts for the LLM. The MILP code are sampled uniformly at random from the successful MILP code generated from the previous level. For an existing code, we select a prompt type by randomly sampling based on the default weights *{"Formulation_Add": 1, "Topic_Add": 0.5, "conv_add": 0.5, "Cross_Over": 1, "Mutate": 1, "Formulation_Mutate": 0.8, "Mutate_redundancy": 0.8, "Topic_new": 1, "New": 0.8, "Delete": 0.5}*, where we lower the weight of certain prompts to balance the amount of prompts from different categories (**Add**, **Mutate**, **Cross-Over**, **New**, **Delete**). If the MILP instance associated with the existing code has a long solve time ($> 150s$), we increase the weight of the Delete prompt to $0.8$ and decrease the weight of all Add prompts to half of the default weights.

We provide a detailed description of each prompt type as follows.

**Prompt Structure** We use chain-of-thought prompting technique to prompt the LLM to generate a new MILP code from a given MILP code. The prompts generally contain the following main components:

1. Summarize the given MILP code.

2. Describe how to modify the MILP code based on the specific prompt type.

3. Generate the new MILP code, step-by-step following a prompt-type specific requirements and a general requirement.

In addition, we give a specific requirement of the generation output format to follow the Data, Optimization Modeling, and Parameters modular structure, and provide the LLM the given MILP code. In the second component, we ask the LLM to describe necessary changes in the new MILP based on different prompt subcategories. Specifically, we have

1. **Formulation Add:** We randomly select three formulation methods within the following list of commonly used MILP formulation methods: *Knapsack Constraints (Set Packing, Set Covering, Set Partitioning), Clique Inequalities, Big $M$ Formulation, Convex Hull Formulation, Logical Conditions, Piecewise Linear Functions, Symmetry Breaking, Special Ordered Sets, Indicator Constraints, Semi-Continuous Variables, Stochastic and Robust Optimization, Network Flow Models*. We ask the LLM to select a formulation method and describe how this method can be applied to the given MILP enhancing the complexity and realism of the MILP formulation.

2. **Topic Add:** We randomly select a topic from a large pool of LLM-generated optimization topics covering various real-world applications and optimization methodologies (see Appendix A.1.3

below). We then prompt the LLM to describe how it can retain the original MILP's structure while adding complexity by incorporating the specific topic into the MILP.

3. **Conversation Add:** We randomly select a topic similar to topic add, but we ask the LLM to generate a dialogue between an expert (assistant) and a novice (user) in the MILP domain around the given topic. We then ask the LLM to summarize the dialogue into a new MILP.

4. **Cross-Over:** Given two existing MILPs, we ask the LLM to generate a new MILP by incorporating information from the second MILP to the second MILP. Specifically, we ask the LLM to embed the two MILPs within the same specific real-world application, explain similarities and differences between the two MILPs, and incorporate the second MILP code to the first code.

5. **General Mutate:** We ask the LLM to describe the MILP under a different real-world scenario and discuss how to the given MILP code can be slightly modified to suite the different real-world application while maintaining a similar level of complexity as the given MILP.

6. **Formulation Mutate:** Similar to Formulation Add, we provide the LLM with three randomly selected formulation methods and ask the LLM to choose a MILP formulation and explain how it can replace some of the existing constraints in the given MILP code by new constraints using the chosen MILP formulation method in the real-world context.

7. **Mutate Remove Redundancy:** We ask the LLM to identify and remove redundancies from the given MILP code and further introduce novel, more diverse component to create the new MILP.

8. **New:** We ask the LLM to describe how the new MILP can follow a similar python syntax and structure as the given MILP but models a completely different optimization problem with a different real-world application.

9. **Topic New:** Similar to Topic Add, we randomly select a topic from the topic pool and ask the LLM to provide a detailed description of new MILP code to suite a different optimization problem under the selected topic with a specific real world application.

10. **Delete:** We ask the LLM to identify less important components from the given MILP and explain how the new MILP can remove the less important components.

The complete prompts of **Formulation Add**, **Cross-Over**, **Mutate**, **Topic New**, **Delete** can be found in Appendix A.4.2.

For the third component, the prompt-type-specific requirements guide the generation (or removal, in the case of deletion) of the data, optimization models, and parameter blocks in the MILP code, with slight wording variations depending on the type. The general requirement outlines instructions on the completeness, modularity, executability, novelty, and difficulty of the generated code. The type-specific requirements are detailed in Appendix A.4.3, while the general requirement is provided in Appendix A.4.4.

### A.1.3  GENERATING MILP PROBLEM TOPICS

To generate MILP topics, we begin with the generation of two distinct sets of topics with GPT-4. The first set, referred to as the *application topics*, focuses on different sectors of the real world. These sectors include logistics, supply chain, healthcare, environmental planning, among others. Each application topic is accompanied by a detailed description, a list of notable companies, associated sub-domains, and various challenges faced within that sector. The second set, *methodology topics*, consists a variety of mathematical models and optimization techniques used in operations research. These include topics like linear programming, Just-In-Time (JIT) models, lean systems, Markov processes, and stochastic programming. Each methodology represents a distinct approach or toolset that can be applied to the application domains to address optimization problems.

Then, we create a MILP topics set – a cross product of the above two sets, where one element from the application set is paired with one element from the methodology set. This process results in the generation of potential MILP topics, each representing a combination of an application sector and a methodology, forming the basis for more specific research or study. In general, we applied GPT-4o to generate 90 companies (from 23 industries) as application topics and 108 methodology topics, which results in 9720 (seed) MILP topics, and then we randomly selected 5000 of them for future tasks.

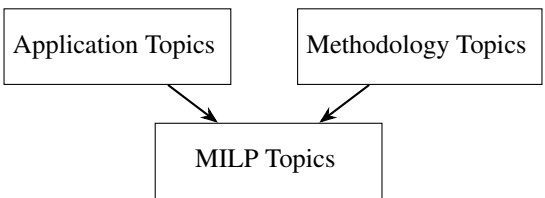

Figure 8: MILP Topic Generation Process.

### A.1.4  Parameter Adjustment and Filtering Details

**Parameter Grid Search Ranges** : For each parameter in the set, we independently sample its value uniformly at random from the following ranges:

- For an integer parameter $v$, we define the search space as $int(v \times [0.5, 0.75, 1, 2, 3, 5, 7, 9, 10, 15])$.
- For a floating point parameter $v$ with the current parameter value $v \geq 1$, we define the search space as $v \times [0.5, 0.75, 1, 2, 3, 5, 7, 9, 10, 15]$.
- For a floating point parameter with the current parameter value $v < 1$, we define the search space as the linear space between $0.1$ and $0.8$ with $11$ equally distance values: $[0.1, 0.17, 0.24, 0.31, 0.38, 0.45, 0.52, 0.59, 0.66, 0.73, 0.8]$.
- For a boolean parameter, we define the search space as [True, False].

**Filtering Criteria:**  given a set of parameter values from grid search, we declare success for the parameter if the following hold

- Solve time $t \in [t_{low}, t_{high}] = [20s, 180s]$
- Presolve Time: $t^{pre} \in [t_{low}^{pre}, t_{high}^{pre}] = [0s, 15s]$, and $\frac{t^{pre}}{t} \in [frac_{low}^{pre}, frac_{high}^{pre}] = [0, 0.2]$
- Total Var $n^{var} \in [n_{low}^{var}, n_{high}^{var}] = [50, 5 \times 10^4]$
- Total Binary and Integer Var: $n^{bin\_int\_var} \in [n_{low}^{bin\_int\_var}, n_{high}^{bin\_int\_var}] = [50, 2 \times 10^4]$
- Total Cons: $n^{cons} \in [n_{low}^{cons}, n_{high}^{cons}] = [50, 5 \times 10^4]$
- Number of branch and bound nodes $n^{bnb} \in [n_{low}^{bnb}, n_{high}^{bnb}] = [10, 5000]$
- Integrality Gap: $gap \in [gap_{low}, gap_{high}] = [0, 300\%]$

The above information can be parsed and computed from the MILP solving log file for the MILP instance associated with the MILP class, given the default seed parameter.

### A.1.5  MILP Instance and Learning Dataset Collection Details

***MILP-Evolve.***  We take the first 800 MILP classes generated by *MILP-Evolve* and generate multiple MILP instances per class using different random seeds, following standard practice from previous studies (Prouvost et al., 2020; Scavuzzo et al., 2022). Details of the instance generation and dataset collection are as follows:

- **Integrality Gap Prediction:** we split the first 800 MILP classes into 643 classes for training/validation and 157 classes for testing ( $\sim$ 8:2 ratio). We generate 100 instances per class, and further split all training/validation instances with a 8:2 ratio into a separate training and validation set.

  To collect the Integrality Gap data, we solve each instance with a time limit of $200s$ and exclude any instance not solved to optimal within the time limit. Among all the optimally solved instances, we clip the integrality gap to the range $[0\%, 100\%]$, where we set $100\%$ to be the upper bound and represents instances with particularly loose LP relaxations. This results in a set of $38,256$ training instances, $9,564$ validation instances and $11,584$ test instances. Data collection with 50 CPUs takes around 66 hours (2.75 days).

- **Learning to Branch:** We split the first 800 MILP classes with roughtly a 7:1:2 ratio into 579 for training, 59 for validation and 162 for testing. We then collect 50, 10 and 30 instances per class for training, validation, and testing. Following the same data collection procedure in Gasse et al. (2019), each B&B node has a probability of 0.95 to apply a Pseudocost-branching strategy for exploration and 0.05 to use the Strong Branching expert to collect the training data. Up to 50 data per instance are collected.

  We set a solve time limit of $200s$ to collect the strong branching data for each instance, excluding instances solved optimally at the root node by default SCIP. This results in a set of $26,502$ MILP instances for training, $512$ instances for validation and $4,756$ instances for testing. Data collection using 50 CPUs takes around 34 hours (1.4 days). Additional instances are collected for baseline models to match the total training instances in Fig. 4.

- **Language-MILP Contrastive Learning:** We generated 10 instances for each of the 1,260 classes, except in the ablation experiment shown in Figure 4c, where we generated more instances for the selected classes. Extracting numerical information from the MPS file only takes a negligible amount of time, and most of the data generation time occurs in querying LLMs. We used 80% of the classes for training and the rest for testing. During training, 10% of the training instances were held out for validation.

**Seed, Seed+Param. and Seed+VAE.**

- **Seed.** For each of the 8 seed classes, we manually select two parameters, resulting in a total of 16 seed parameters. We then collect the learning datasets to train the **Seed** model in Table 1:

  (a) **Integrality Gap Prediction:** We solve 100 MILP instances for each parameter, with a time limit of $200s$ per instance. We then split the optimally solved instances into a set of 1208 training instances and a set of 303 validation instances ($\sim 8{:}2$ ratio).

  (b) **Learning to branch:** We solve 500 MILP instances per parameter, using the same 200-second time limit, resulting in 7252 training instances. For validation, we use the same MILP-Evolve set without further splitting the seed instances.

  (c) **Language-MILP Contrastive Learning:** We generated $1,446$ instances and used 20% instances as a validation set to select the parameters (learning rate, dropout, etc.); then, we used the entire dataset for training.

- **Seed + Param.** For each of the 8 seed classes, we use our parameter search and filtering procedure in *MILP-Evolve* (Fig. 2, Appendix A.1.4) to generate additional valid parameters. We sample 240 parameters per seed class, from which we obtain **89 new parameters** that satisfy the filtering criteria. Then we collect the learning dataset for the new parameters to augment the dataset of the Seed Classes, which we use to train the **Seed + Param.** model in Table 1. Specifically,

  (a) **Integrality Gap Prediction:** We generate 100 MILP instances for each new parameter, resulting in 7462 augmented MILP instances with Integrality Gap values. These are combined with the seed dataset, resulting in a final set of 7177 training instances and 1796 validation instances.

  (b) **Learning to Branch:** We further collect the strong branching data for 100 instances per parameter, excluding instances solved optimally at the root node by SCIP Default. This yields a total of 13831 training instances. We similarly use the same MILP-Evolve set for validation.

  (c) **Language-MILP Contrastive Learning:** We generated $7,389$ instances and used 20% instances as a validation set to select the parameters (learning rate, dropout, etc.); then, we used the entire dataset for training.

- **Seed + VAE.** We adopt the state-of-the-art instance generation method from Guo et al. (2024) to augment the seed MILP instances. Specifically, we first train a Variational Autoencoder (VAE)-based model on the combined set of Seed training and validation instances. The trained VAE generates $21,000$ new instances, with $12000, 6000, 3000$ instances generated for mask ratios of $\eta = \{0.01, 0.05, 0.1\}$ (the fraction of constraints modified). We then collect datasets for the **Seed + VAE** model in Table 1. We note that we find many of the generated instances are infeasible and cannot be used for training.

  (a) **Integrality Gap Prediction:** We obtain 8316 valid instances. Combined with the seed dataset, this results in a final set of 7860 training and 1967 validation instances.

(b) **Learning to Branch:** We collect strong branching data for 7552 instances. Combined with the seed dataset, this provides a total of $14,804$ training instances. We again use the same MILP-Evolve set for validation.

(c) **Language-MILP Contrastive Learning:** We generated $8,305$ instances and used 20% instances as a validation set to select the parameters (learning rate, dropout, etc.); then, we used the entire dataset for training.

**Language-MILP Contrastive Learning: Description Collection.** To generate a meaningful description of an MILP problem from its MILP instance file (where we use the MPS format (Tomlin & Welch, 1992) to save the instances), we combine both the characteristics extracted from the source code and the values from the MPS file to query a LLM to produce an accurate textual description that can capture both the high-level problem information and data information from the A/B matrices. **Generating Problem Characteristics:** The process begins by using a Large Language Model (LLM) to extract specific characteristics of the problem from the solver's Python source code (which can be written using SCIP, Pyomo, or Gurobi). The LLM analyzes the solver code, identifying key attributes such as the MPS format details, formulation techniques (e.g., "big M" or inequality formulations), problem domain (e.g., "bin packing" or "set cover"), as well as important details about the objective function, constraints (linear or non-linear), and variable types (integer, binary, continuous). **Extracting MPS Information:** After identifying problem characteristics, a rule-based algorithm parses the MPS file into its key sections: ROWS, COLUMNS, RHS, and BOUNDS. Each of these sections corresponds to different aspects of the MILP problem. For example, ROWS defines the objective function and constraints, COLUMNS identifies the coefficients of the decision variables, RHS contains the right-hand side values for the constraints, and BOUNDS specifies the variable bounds.

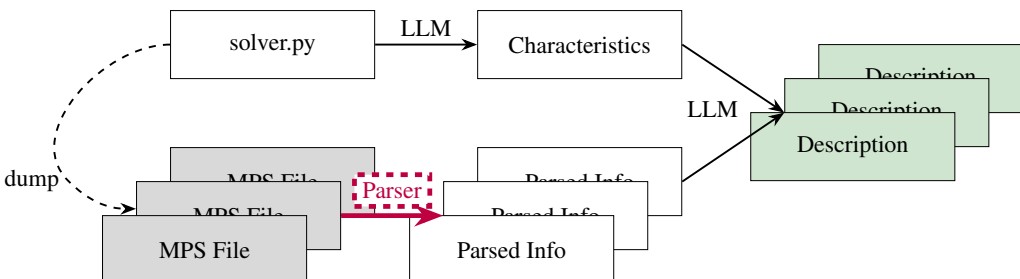

Figure 9: Process for generating descriptive text from MILP problems using source code and MILP instance files (MPS format). The generated text is then used for the language-MILP contrastive learning task.

### A.1.6 MIPLIB Dataset Collection Details

MIPLIB (Gleixner et al., 2021) is a widely used benchmark dataset for MILP, featuring a diverse range of instances from various application domains. We study transfer learning on this heterogeneous dataset for Integrality Gap Prediction and Language-MILP Contrastive Learning.

**Integrality Gap Prediction** A majority of MIPLIB instances (specifically, $914$ of them) have their optimal solution values posted on the website. We curate this set of instances and compute the integrality gap based on the posted optimal solution. We split the set into $614$ instances for training (fine-tuning) and $300$ instances for testing for the MIPLIB experiments in Table 3

**Language-MILP Contrastive Learning.** Each MIPLIB instance is given as the $A, b$ matrices rather than the optimization problem formulation (e.g. constraints and variables that can be used to generate descriptions). However, most instances include a brief description. A subset of these descriptions are informative and suitable for the language task. For example, the description of the 'map10' instance is: 'Land parcel selection problems motivated by Red-Cockaded Woodpecker conservation problem Imported from MIPLIB2010[1]. In contrast, other descriptions are less informative

---
[1]https://miplib.zib.de/instance_details_map10.html

and not useful for the language task. For instance, the 'neos-1582420' instance is described simply as 'Collection of anonymous submissions to the NEOS Server for Optimization[2].

To ensure that only relevant and informative instances were retained, we applied a large language model (LLM) to filter out instances with descriptions deemed uninformative. As a result, the dataset was reduced from an initial set of 914 instances for Integrality Gap Prediction to a final count of 303 instances. We then divided the dataset with a 8:2 ratio into a training (fine-tuning) set of 242 instances and and a test set of 61 instances. This division provided a sufficient balance between training the model and retaining a subset for unbiased evaluation.

Given the relatively small size of the filtered dataset, we conducted the testing process 10 times and averaged the results to ensure robustness. For Figure 4c, where only 1,000 instances were used for training, we repeated the training process four times with different randomly selected data, performing 10 rounds of testing each time, and reported the average. This repetition mitigated any potential variability that could arise from the limited number of instances.

**Data filtering details.** Examples of the descriptions that were filtered out include those related to undisclosed industrial applications from companies like Google, instances imported from earlier MIPLIB submissions, or those collected from forums such as the Gurobi forum with unknown applications. Some instances were also removed because they were marked as infeasible by optimization solvers such as ParaSCIP, taking an extended time to solve, or because they were anonymous submissions to optimization servers like NEOS. Other filtered instances originated from MiniZinc Challenges between 2012 and 2016 or were randomly generated integer and binary programming instances. Descriptions related to railway line planning and other irrelevant application contexts were also excluded.

We report the MPS-to-Text accuracies in our results because we only fine-tuned the GNN model and froze the text embedding model. Interestingly, we observed that the Text-to-MPS accuracy was consistently about 1–2% higher than the MPS-to-Text accuracy, suggesting that the text encoder in our CLIP model was better trained than the graph neural network (GNN) encoder used for the Mixed-Integer Programming (MIP) instances.

## A.2 MILP Learning Details

### A.2.1 Integrality Gap Prediction

**Training and Evaluation Setup.** We train, validate and test all methods on a distributed cluster using nodes equipped with 80 Intel(R) Xeon(R) Silver 4316 CPU and A single Nvidia Volta A100 GPU. Training takes less than 24 hours for each model.

*Training Loss.* At training time, we use Huber Loss (Huber, 1992) to minimize the deviation of predicted integrality gap from the label. Specifically, given a ground truth label $g^*(x)$ and a predicted integrality gap $\hat{f}_{theta}(x)$ for a MILP instance $x$, the Huber Loss is defined as

$$L(\theta; x) = \begin{cases} \frac{1}{2}(\hat{f}_\theta(x) - g^*(x))^2, & \text{if } |\hat{f}_\theta(x) - g^*(x)| \leq 1 \\ |\hat{f}_\theta(x) - g^*(x)| - \frac{1}{2}, & \text{otherwise.} \end{cases} \tag{2}$$

*Evaluation Metric.* At test time, we report the absolute deviation $\frac{1}{|\mathcal{X}_{\text{test}}|} \sum_{x \in \mathcal{X}_{\text{test}}} |\hat{f}_\theta(x) - g^*(x)|$ across a set of multi-class test instances $x \in \mathcal{X}_{\text{test}}$. We further evaluate the Pearson Correlation of the prediction and the ground truth as the second test metric.

**Input Features** We use the variable and constraint features provided in Paulus et al. (2022). A list of the features are provided in Table 5. As we only need to extract the state information after the first root-node LP relaxation, we remove a subset of irrelevant constraint features related to cutting planes from the original set.

Note: We use the Paulus et al. (2022) implementation instead of Ecole (Gasse et al., 2019) because the customized PySCIPOpt interface by Paulus et al. (2022) allows us to extract input features after

---

[2]https://miplib.zib.de/instance_details_neos-1582420.html

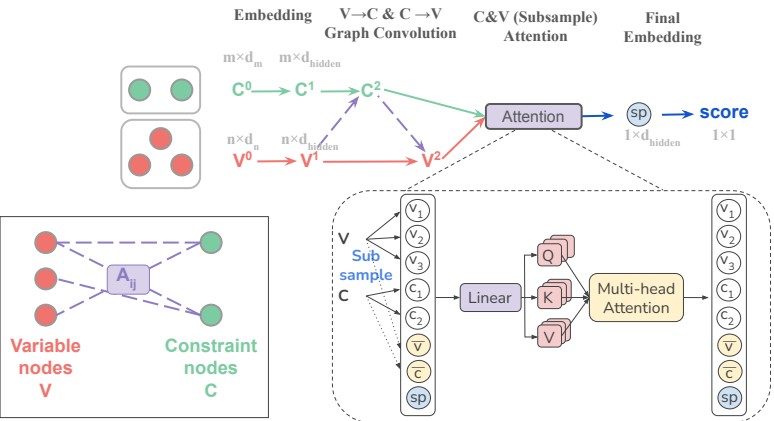

Figure 10: Input Graph Representation and Learning Architecture for Integrality Gap Prediction. For *Ours* - Attn., the attention layer is replaced with global pooling on the hidden embeddings of all constraints and variables to obtain the global summary vector $sp$.

each LP relaxation, whereas that by Gasse et al. (2019) only allows extracting input features at each branching decisions, and the first branching decision happens later than the first LP relaxation.

Table 5: Integrality Gap Prediction and Language-MILP Contrastive Learning: MILP instance input features for variable and constraint nodes (Paulus et al. (2022)).

| Node Type | Feature | Description |
|---|---|---|
| **Vars** | norm coef | Objective coefficient, normalized by objective norm |
| | type | Type (binary, integer, impl. integer, continuous) one-hot |
| | has lb | Lower bound indicator |
| | has ub | Upper bound indicator |
| | norm redcost | Reduced cost, normalized by objective norm |
| | solval | Solution value |
| | solfrac | Solution value fractionality |
| | sol_is_at_lb | Solution value equals lower bound |
| | sol_is_at_ub | Solution value equals upper bound |
| | norm_age | LP age, normalized by total number of solved LPs |
| | basestat | Simplex basis status (lower, basic, upper, zero) one-hot |
| **Cons** | rank | Rank of a row |
| | norm_nnzrs | Fraction of nonzero entries |
| | bias | Unshifted side normalized by row norm |
| | row_is_at_lhs | Row value equals left hand side |
| | row_is_at_rhs | Row value equals right hand side |
| | dualsol | Dual LP solution of a row, normalized by row and objective norm |
| | basestat | Basis status of a row in the LP solution, one-hot |
| | norm_age | Age of row, normalized by total number of solved LPs |
| | norm_nlp_creation | LPs since the row has been created, normalized |
| | norm_intcols | Fraction of integral columns in the row |
| | is_integral | Activity of the row is always integral in a feasible solution |
| | is_removable | Row is removable from the LP |
| | is_in_lp | Row is member of current LP |
| | obj_par | Objective parallelism score of a row |

**Architecture and Training Hyperparameters.** Our network, as illustrated in Fig. 10, first embeds $\mathbf{V} \in \mathbb{R}^{n \times 17}$, $\mathbf{C} \in \mathbb{R}^{m \times 29}$ (where $n, m$ are the number of variables and constraints) into hidden representations of dimension $d_{hidden} = 64$ with a BatchNorm followed by two (Linear, ReLU)

Table 6: **Architecture hyperparameters for Integrality Gap Prediction.**

Table 7: **Training hyperparameters.**

| Input dimension $n \times d_n$ $m \times d_n$ | $\mathbf{V} \in \mathbb{R}^{n \times 17}$ $\mathbf{C} \in \mathbb{R}^{m \times 29}$ | GCN Message Passing Order | $\mathbf{V} \rightarrow \mathbf{C}$, $\mathbf{C} \rightarrow \mathbf{V}$ |
|---|---|---|---|
| Output dimension | 1 | Num. GCN Layers | 1 |
| Embedding dimension $d_{hidden}$ | 64 | Attention Num. Heads | 8 |
| Num. sampled Cons.& Vars. $s_{subsample}$ | 512 | Attention Dropout | 0.6 |

| Optimizer | Adam |
|---|---|
| Learning rate | 0.001 |
| Batch size | 32 |
| Num. of Gradient Steps | 30000 |

blocks. The hidden embeddings are fed into a Graph Convolution module Kipf & Welling (2017), with message passing, following the direction of $\mathbf{V} \rightarrow \mathbf{C}$ and $\mathbf{C} \rightarrow \mathbf{V}$, with a final (LayerNorm, Linear, ReLU, Linear) block that maintains the dimension $d_{hidden}$. Then, for each MILP instance, we randomly sample a subset of $s_{subsample} = 512$ constraint and variable nodes and construct a $(s_{subsample} + 3) \times d_{hidden} = 515 \times 64$ matrix, where the first $s_{subsample}$ dimensions are the subsampled constraint and variable hidden embedding, and the last three dimensions are the mean constraint embedding, mean variable embedding, and a special summary node where we extract the global information to. Lastly, we feed this matrix into a Transformer Encoder Layer Vaswani (2017) with $n_{heads} = 8$ heads and a $dropout$ rate of 0.6, and we take the attention output embedding of the all-zero input vector and finally use a (Linear, ReLU, Linear) block to map the vector into a scalar output.

We train with Adam optimizer with a learning rate of 0.001 and a batch size of 32 with a total of 30000 gradient steps. All hyperparameters are selected on the validation set and frozen before evaluating on the test set. Table 6 and 7 provides a list of hyperparameters.

### A.2.2 LEARNING TO BRANCH

**B&B background and common branching strategies.** Exact MILP solvers (Bestuzheva et al., 2021; Gurobi Optimization, LLC, 2023) typically employ the *Branch-and-Bound* (B&B) algorithm, which systematically explores a search tree to find the optimal solution. At each node in the tree, a Linear Programming (LP) relaxation is solved, where the integrality constraints on the variables are relaxed. If the LP solution satisfies the integrality constraints, it is feasible for the MILP, and the process can backtrack. Otherwise, the algorithm selects a variable to branch on, creating two child nodes with additional constraints (e.g., $x_j \leq \lfloor x_j^* \rfloor$ and $x_j \geq \lceil x_j^* \rceil$).

The efficiency of the B&B algorithm heavily depends on the *branching strategy* used to select variables for branching (Achterberg, 2007). Common strategies include:

- **Most Infeasible Branching**: Selecting the variable with the value closest to fractional (i.e., $x_j^*$ closest to 0.5).

- **Strong Branching**: Evaluating the potential impact of branching on each candidate variable by temporarily branching and estimating the resulting lower bounds.

- **Pseudo-Cost Branching**: Estimating the effect of branching based on historical information gathered during the search.

However, these heuristics may not be optimal for all problem instances, and designing effective branching strategies remains an area of active research. In this work, we extend the setup of (Gasse et al., 2019) to imitate the Strong Branching expert to the multi-class learning context.

**Training and Evaluation Setup.** We train, validate and test all methods on a distributed cluster using nodes equipped with 80 Intel(R) Xeon(R) Silver 4316 CPU and A single Nvidia Volta A100 GPU. Training for each model takes less than 24 hours. During data collection and testing, we use a single CPU to solve each MILP instance. Following Gasse et al. (2019), we disable presolving and cutting plane separation to focus on the impact of learning for B&B. A $200s$ time limit is set for solving each instance.

*Training Loss.* Each training data corresponds to a state-action pair $(s, a^*)$ at a B&B node, where $s$ represents the MILP subproblem at the node and $a^*$ is the strong branching expert action. Given a learned policy $\hat{f}_\theta(\cdot)$ that outputs the probability of selecting each variable from a candidate set, we minimize the cross-entropy loss $-\frac{1}{N} \sum_{(s,a^*)} \log \hat{f}_\theta(a^*|s)$ of predicting the expert action.

*Evaluation Metric.* At test time, we compare the solve time of each learned method with SCIP Default for each test instance, with a $200s$ solve time limit per instance. For each method, we report the proportion instances solved to optimal, we report the proportion of instances solved to optimal, the 1-shifted geometric mean of time improvement over SCIP Default, and the 1-shifted geometric mean time improvement excluding neural network overhead (Ex.). Time improvements are calculated for instances where the learned method solves to optimal within the time limit.

**Input Features** We follow the Ecole (Gasse et al., 2019) implementation for the learning to branch experiments. As listed in Table 8, the input features are similar to those in Paulus et al. (2022) with a slightly larger set of variable features and a smaller set of constraint features.

**Architecture and Training Hyperparameters.** We adopt a similar GCN architecture as in Gasse et al. (2019). Specifically, given the Ecole input feature with $\mathbf{V} \in \mathbb{R}^{n \times 19}$ and $\mathbf{C} \in \mathbb{R}^{m \times 5}$, the model includes the same embedding layer as in Fig. 10, followed by three layers of the $\mathbf{V} \rightarrow \mathbf{C}$ & $\mathbf{C} \rightarrow \mathbf{V}$ Graph Convolution. We increase the number of layers as it improves the prediction accuracy without adding significant overhead. After the GCN block, each variable node's hidden embedding is projected through an MLP; this results in a predicted score for each variable, which we use to select the variable. We set the same hidden dimension of $n_{hidden} = 64$ and train each model for 100 epochs. The remaining hyperparameters are identical to those in Table 6 and 7.

Notably, we exclude the attention layer from the architecture due to its computational overhead: since we predict a score for each variable rather than for the whole graph, subsampled attention from Fig. 10 cannot be applied, and using full attention for all variables would increase the solve

Table 8: Learning to Branch: MILP instance input features for variable and constraint nodes (Gasse et al., 2019).

| Node Type | Feature | Description |
|---|---|---|
| **Vars** | norm coef | Objective coefficient, normalized by objective norm |
| | type | Type (binary, integer, impl. integer, continuous) one-hot |
| | has lb | Lower bound indicator |
| | has ub | Upper bound indicator |
| | norm redcost | Reduced cost, normalized by objective norm |
| | solval | Solution value |
| | solfrac | Solution value fractionality |
| | sol_is_at_lb | Solution value equals lower bound |
| | sol_is_at_ub | Solution value equals upper bound |
| | norm_age | LP age, normalized by total number of solved LPs |
| | incumbent_value | The objective value of the current best solution |
| | avg_incumbent_value | he mean of all incumbent values found so far |
| | basestat | Simplex basis status (lower, basic, upper, zero) one-hot |
| **Cons** | bias | Unshifted side normalized by row norm |
| | obj_cosine_sim | Cosine Similarity of the row with the objective |
| | is_tight | Row value equals right hand side |
| | dualsol | Dual LP solution of a row, normalized by row and objective norm |
| | norm_age | Age of row, normalized by total number of solved LPs |

time (our main evaluation criterion). We recognize developing an efficient attention mechanism to enhance multi-class branching as an important direction for future work.

### A.2.3 MILP-LANGUAGE CONTRASTIVE LEARNING

**Loss Details.** Let $\mathcal{D} = \{(M_i, T_i)\}_{i=1}^{N}$ denote a dataset comprising $N$ pairs of MILP instances $M_i$ and their corresponding textual descriptions $T_i$. Our objective is to learn embedding functions $f_M : \mathcal{M} \to \mathbb{R}^d$ and $f_T : \mathcal{T} \to \mathbb{R}^d$ that map MILP instances and text descriptions into a shared $d$-dimensional latent space.

For a batch of $K$ MILP-text pairs, we compute the embeddings:

$$
\begin{aligned}
\mathbf{h}_M &= f_M(M) \in \mathbb{R}^{K \times d}, \\
\mathbf{h}_T &= f_T(T) \in \mathbb{R}^{K \times d},
\end{aligned}
\tag{3}
$$

where $\mathbf{h}_M$ and $\mathbf{h}_T$ are the embeddings of the MILP instances and text descriptions in the batch, respectively.

We then normalize these embeddings to have unit length:

$$
\begin{aligned}
\tilde{\mathbf{h}}_M &= \text{normalize}(\mathbf{h}_M), \\
\tilde{\mathbf{h}}_T &= \text{normalize}(\mathbf{h}_T),
\end{aligned}
\tag{4}
$$

where the normalization is performed along the embedding dimension, applied row-wise for each example in the batch:

$$
\text{normalize}(\mathbf{h}_i) = \frac{\mathbf{h}_i}{\|\mathbf{h}_i\|_2},
\tag{5}
$$

Next, we compute the similarity scores (logits) between all pairs in the batch using the dot product of the normalized embeddings:

$$
\begin{aligned}
\mathbf{Z}_{M \to T} &= \tilde{\mathbf{h}}_M \tilde{\mathbf{h}}_T^\top \in \mathbb{R}^{K \times K}, \\
\mathbf{Z}_{T \to M} &= \tilde{\mathbf{h}}_T \tilde{\mathbf{h}}_M^\top \in \mathbb{R}^{K \times K}.
\end{aligned}
\tag{6}
$$

Here, $\mathbf{Z}_{M \to T}[i, j]$ represents the cosine similarity between the $i$-th MILP instance and the $j$-th text description.

We define the labels for contrastive learning as:

$$
\mathbf{y} = [0, 1, 2, \ldots, K - 1],
\tag{7}
$$

indicating the correct matching pairs in the batch.

We then compute the cross-entropy losses for both MILP-to-text and text-to-MILP directions:

$$
\begin{aligned}
\mathcal{L}_{M \to T} &= \frac{1}{K} \sum_{i=1}^{K} \text{CrossEntropy}\left(\mathbf{Z}_{M \to T}[i, :], \mathbf{y}[i]\right), \\
\mathcal{L}_{T \to M} &= \frac{1}{K} \sum_{i=1}^{K} \text{CrossEntropy}\left(\mathbf{Z}_{T \to M}[i, :], \mathbf{y}[i]\right).
\end{aligned}
\tag{8}
$$

The total loss is the average of the two losses:

$$
\mathcal{L} = \frac{1}{2}\left(\mathcal{L}_{M \to T} + \mathcal{L}_{T \to M}\right).
\tag{9}
$$

**Training and Evaluation Setup.** We train, validate and test all methods on a distributed cluster using nodes equipped with 80 Intel(R) Xeon(R) Silver 4316 CPU and A single Nvidia Volta A100 GPU. Training takes less than $24hr$ for each model.

*Engineering Tricks.* We used a caching mechanism for the text embedding model because it was not fine-tuned during our experiments. Hence, we only needed to utilize the 7B embedding model once per text description, which largely improved the efficiency of the training process.

*Evaluation Metric.* At test time, given a MILP instance $x$ and a set of natural language descriptions $\{y_1, \ldots, y_k\}$ for different MILP classes, we then perform a $k$-way classification to distinguish the correct natural language description for each MILP instance from the set of options. We report the 4-way and 10-way accuracy on the test set ($k = 4$ and 10).

**Input Features** For the MILP, we use the same input feature as in Table 5 to embed each MILP instance $x$; that is, the input graph is constructed after the first root-node LP is solved, which is easy to obtain as the root-node LP is typically fast to solve. The text description is fed directly into a language encoder.

**Architecture and Training Hyperparameters.** For the MILP encoder, we adopt the same architecture as Integrality Gap Prediction (Fig. 10, Table 6) with change in the final layer (head), instead of projecting into a single score, we project into a embedding vector with dimension $n^{output} = 4096$ so that we can project the modality between MILP and text (NV-Embed-v1[3] with temperature 0). We set the model with dropout ratio 0.5 and train the model with learning rate $5 \times 10^{-5}$ with Adam Optimizer for 100 epochs.

---

[3] https://huggingface.co/nvidia/NV-Embed-v1

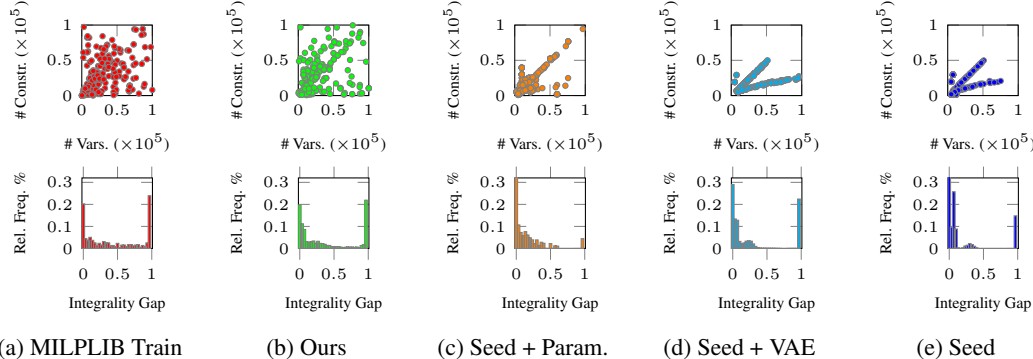

(a) MILPLIB Train     (b) Ours     (c) Seed + Param.     (d) Seed + VAE     (e) Seed

Figure 11: MILP Instance Statistics from Different Data Sources. We plot the problem size distribution (top) and Integrality Gap training label distribution (bottom). Instances generated by *MILP-Evolve*(*Ours*) exhibits a closer problem size label distribution to MIPLIB than baseline methods. See Appendix A.3.1 for details.

## A.3 ADDITIONAL EXPERIMENTAL RESULTS AND DISCUSSIONS

### A.3.1 MILP INSTANCES STRUCTURAL ANALYSIS

In Table 9, we present structural statistics to compare the different data sources (*MILP-Evolve*, Seed, VAE(Guo et al., 2024), Param. Search). We also include statistics from the MIPLIB training set (Sec. 5.3) to understand why *Ours* outperforms methods trained on Seed instances or their augmented versions in the transfer learning setting. The statistics are categorized into four groups, each representing a different aspect of MILP structure. We visualize a subset of these statistics in Fig. 11.

1. **Class**: We compare the optimization problem formulations of classes generated by *MILP-Evolve* with the 8 seed classes. We extract the number of `model.addVar` and `model.addCons` statements in each code, which correspond to different types of variables and constraints. As seen in Fig.7, MILP classes generated by *MILP-Evolve* include a wider variety of variable and constraint types. We note that this aligns with real-world optimization problems, which typically involve many types of variables and constraints to capture the complexity of the problem.

2. **Instances**: We analyze the structural properties of MILP instances generated from different data sources, including instance size (number of variables and constraints), the ratio of continuous variables, and constraint coefficient density (proportion of non-zero entries in the $A$ matrix). While the continuous variable ratio and constraint densities are similar across sources, instances from *MILP-Evolve* have more variables and constraints than **Seed**, **VAE**, and **Param**. This is expected, as *MILP-Evolve* generates more diverse MILP formulations. As shown in Fig. 11, the *MILP-Evolve* produces instances with a diverse range of sizes, similar as the MIPLIB dataset; in contrast, **VAE** and **Param.** can only perturb instances slightly from the Seed instances.

3. **Solving**: We measure the distribution of solve time and the number of B&B nodes across different data sources. Fig. 12 further shows the solve time distribution for **Seed**, **VAE**, **Param.**, and *MILP-Evolve*. We observe that instances generated by *MILP-Evolve* have similar solve times to **Seed**. In contrast, **VAE** instances have shorter solve times, which highlights the challenge of preserving solve times in learning-based instance generation. We note that the parameter search and filtering in *MILP-Evolve* is key to maintaining similar solve times to **Seed**; without it, generated instances typically solve in under 5 seconds. This demonstrates the importance of post-filtering to maintain desired problem properties for LLM-based MILP class generation.

4. **Integrality Gap similarity with MIPLIB Test**: We compute how similar each gap distribution from different data sources are to MIPLIB Test. We find that the gap distribution from *MILP-Evolve* is closer to MIPLIB Test than the baseline methods (Seed, VAE, and Param.). As MIPLIB contains a set of heterogeneous instances with a diverse set of gap distribution, this highlights the benefit of *MILP-Evolve* in increasing training instance label diversities, which we see in Table 3 leads to an improved transfer learning performance.

Table 9: Instance Statistics from Different Data Sources.

| | | MIPLIB Train | MILP-Evolve | VAE | Param. | Seed |
|---|---|---|---|---|---|---|
| **Class** | # model.addVar | N/A | 4.28 ±2.66 | 1.58 ± 0.67 | 1.58 ± 0.67 | 1.58 ± 0.67 |
| | # model.addCons | N/A | 6.19 ± 3.46 | 1.84 ± 0.87 | 1.84 ± 0.87 | 1.84 ± 0.87 |
| **Instance** | # Variables | 15666 ± 37561 | 16039 ± 21235 | 3443 ± 3579 | 5350 ± 9336 | 3592 ± 5064 |
| | Ratio of Continuous Vars | 0.65 ± 0.37 | 0.70 ± 0.40 | 0.71 ± 0.44 | 0.83 ± 0.37 | 0.75 ± 0.43 |
| | # Constraints | 14514 ± 41100 | 12367 ± 22461 | 2534 ± 3402 | 6522 ± 11886 | 3087 ± 5064 |
| | Constraint Coefficient Density | 0.040 ± 0.145 | 0.013 ± 0.082 | 0.014 ± 0.020 | 0.041 ± 0.07 | 0.017 ± 0.022 |
| **Solving** | Number of nodes | N/A | 926 ± 4668 | 443 ± 869 | 1454 ± 4965 | 1847 ± 5765 |
| | Pre-solve Time (s) | N/A | 1.82 ± 6.80 | 0.14 ± 0.27 | 0.68 ± 0.89 | 0.22 ± 0.43 |
| | Solve Time (s) | N/A | 56.78 ± 49.19 | 27.88 ± 35.66 | 59.92 ± 45.58 | 59.02 ± 48.74 |
| **Integrality Gap Similarity w/ MIPLIB Test** | Correlation ($\uparrow$) | 0.98 | 0.92 | 0.86 | 0.56 | 0.65 |
| | Intersection ($\uparrow$) | 0.88 | 0.81 | 0.67 | 0.56 | 0.51 |
| | Chi-Square Dist. ($\downarrow$) | 0.07 | 0.13 | 0.39 | 0.52 | 0.64 |
| | Bhattacharyya dist. ($\downarrow$) | 0.02 | 0.04 | 0.15 | 0.21 | 0.28 |

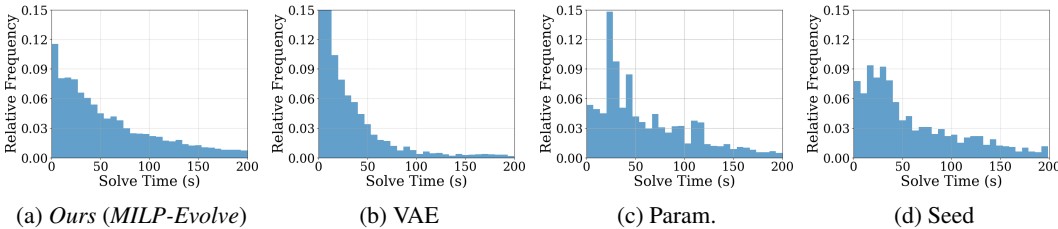

(a) *Ours* (*MILP-Evolve*)   (b) VAE   (c) Param.   (d) Seed

Figure 12: Solve Time Distribution from Different Data Sources. The solve times for *Ours* (*MILP-Evolve*), Seed, and Param. Search are similar, while VAE-augmented instances have slightly shorter solve times.

**Similarity Metric Details.** For each data source (including MIPLIB Test), we first construct a histogram with values between $[0, 1]$ using $n_{bins} = 30$. The histogram for different data sources are visualized in Fig. 11; the MIPLIB Test distribution (not plotted) is similar to MIPLIB Train. We then calculate the similarity of the histogram of each data source and the MIPLIB Test histogram using similarity measures (i) Correlation (higher the better), which computes the Pearson correlation of two histograms; (ii) Intersection (higher the better), calculated as $\frac{\sum_{i=1}^{n_{bins}} \min(hist1[i], hist2[2])}{\sum_{i=1}^{n_{bins}} \max(hist1[i], hist2[i])}$ (iii) Chi-Square Distance (lower the better), computed as $\sum_{i=1}^{n_{bins}} \frac{(hist1[i] - hist2[i])^2}{hist1[i] + hist2[i] + eps}$ (iv) Bhattacharyya Distance (lower the better), calculated as $-\log(\sum_{i=1}^{n_{bins}} \sqrt{hist1[i] * hist2[i]})$.

### A.3.2 LEARNING TO BRANCH: PERFORMANCE PER CLASS.

In Fig. 13, we show the branching performance on each test class for **Seed** and *Ours*. We calculate the 1-shifted geometric mean of time improvement over SCIP Default for all instances within each MILP class, accounting for neural network overhead. Using the same T-SNE embedding from Fig.7, each test class is represented as a circle. A blue circle indicates the method improves solve time over SCIP Default, while a red circle indicates a slower solve time (with darker shades representing greater improvement or degradation). **Seed** is trained only on the seed classes, plotted as orange stars, while *Ours* is trained on both the seed classes and the *MILP-Evolve* generated classes, shown as light gray stars. We observe that *Ours* improves solve time for significantly more classes than **Seed**, though there is still a subset where *Ours* underperforms compared to SCIP Default. This suggests future work on enhancing multi-class learning performance for these classes, or developing a meta-model to select between SCIP Default and a learned method based on problem characteristics.

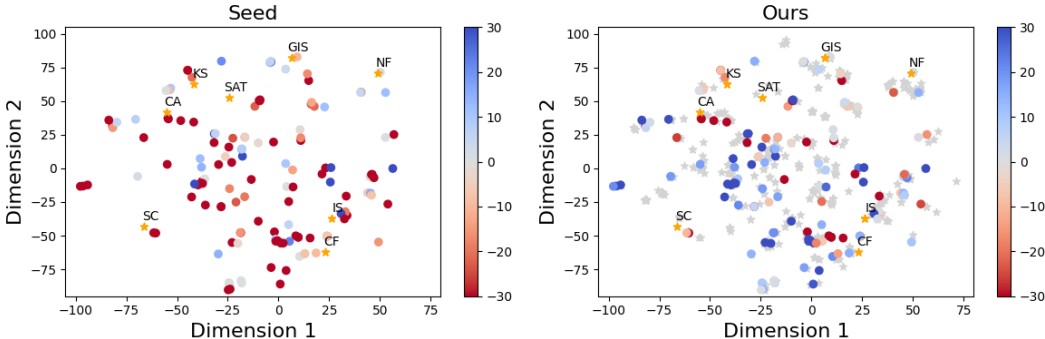

Figure 13: **Learning to Branch Solve Time Improvement in Each MILP Class.** Using the same T-SNE embedding as in Fig. 7, we visualize branching performance on individual test classes. Each class is represented by a circle: blue indicates improved solve time over SCIP Default, red indicates slower solve time, with darker shades showing greater improvement or degradation. **Seed** is trained on seed classes (orange stars), and *Ours* is trained on both seed and *MILP-Evolve*-generated classes (light gray stars).

### A.3.3  A NEW *MILP-Evolve* SET BASED ON SIX UNSEEN SEED CLASSES.

**Six Unseen Seed Classes.**  We take a different set of 6 unseen seed classes, consisting of Graph Coloring, Job-Shop Scheduling, Protein Folding, Multi-Item Lot Sizing, Bin Packing, and Max Cut. We run *MILP-Evolve* to slightly expand the new test set to a total of 50 classes. We perform transfer learning of different models to this unseen test set, where we use 40% classes for fine-tuning and set aside 60% classes for testing.

Table 10: Abbreviations, Full Names, and References for another set of 6 Unseen Seed Classes.

| Abbreviation | Full Name | Reference |
| --- | --- | --- |
| GC | Graph Coloring | Jensen & Toft (2011) |
| JS | Job-Shop Scheduling | Xiong et al. (2022) |
| PF | ProteinFolding | Williams (2013) |
| LS | Multi-Item Lot Sizing | Chen & Thizy (1990) |
| BP | Bin Packing | Tang et al. (2020) |
| MC | Max Cut | Tang et al. (2020) |

**Language-MILP Contrastive Learning: Expert Curate / Verified Language Labels.**  To ensure the quality of language descriptions, we manually verified and modified the linguistic description and made sure the descriptions matches the optimization problem in the testing set.

We observed that GPT-generated descriptions are generally accurate, correctly reflecting the optimization problem and highlighting potential real-world applications of the optimization class (e.g., "This type of scheduling is essential in manufacturing and project management, where minimizing total completion time across multiple tasks is critical."). Some descriptions included irrelevant details (e.g., "PySCIPOpt is used to solve the optimization problem"), which we manually removed.

### A.3.4  EFFECTS OF DIFFERENT SEED CLASSES.

We include an ablation study to analyze the effect of different seed classes on the performance.

**Statistics.** Table 11 shows the proportion of generated classes from each seed class[4]. We see that some seed classes can lead to more generated classes (e.g. Combinatorial Auction (CA)) than others

---

[4]Note that for cross-over prompts, we only tracked the trace of the first MILP class, so the number here can be slightly noisy.

(e.g. Knapsack (KS)). One potential reason could be that our filtering and parameter adjustment procedures can find good solutions for certain classes more easily than others.

Table 11: Proportion of *MILP-Evolve* generated classes from each of the seed class.

|  | IS | CA | KS | GIS | NF | SC | SAT | CF |
|---|---|---|---|---|---|---|---|---|
| Proportion | 10.8% | 27.3% | 4.3% | 22.4% | 7.5% | 6.5% | 11.9% | 9.3% |

**Impact of different classes on the learning performance.** For each of the eight seed classes, we train separate models on instances (1) from all evolved classes based *only* on this seed class (**One Seed**), (2) with *weighted sampling*, 70% from all evolved classes based on this seed class, and 30% based on other seed classes (**Weighted**). We fix the number of train and validation instances, and compare the test performance on *MILP-Evolve* held out test set (**Table 1**) and the transfer learning performance to **MIPLIB** (Table 3).

**Results:** We report our findings in Table 12. We see that

1. **Table 1, One Seed**: Learning on classes based on one single seed class has limited performance.

2. **Table 1, Weighted**: Classes with a higher proportion in the *MILP-Evolve* dataset (CA, GIS), when given a higher weight when sampling training set, typically lead to a better test performance on the *MILP-Evolve* held out set; an exception is Capacitated Facility Location (CF), which has a lower ratio than CA and GIS, but its learned model achieves the best test performance among all models.

3. **MIPLIB, Weighted:** The transfer learning performance when initializing with the different models seem to be similar on the MIPLIB test set, and is worse than *Ours* performance when trained on instances from all MILP classes.

Finally, these results give more evidence to the primary hypothesis of this paper: **the importance of having a diverse set of MILP classes from different seed classes to improve the generalization performance.**

Table 12: Generalization performance on the *MILP-Evolve* hold out set (Table 1) and MIPLIB (Table 3) when learning on (1) One Seed: classes originating from a single seed, and (2) Weighted: 70% classes originating from a single seed, 30% classes from other seeds. We fix the total number of training and validation instances across all settings.

|  | Table 1, One Seed | | Table 1, Weighted | | MIPLIB, Weighted | |
|---|---|---|---|---|---|---|
|  | Dev. ($\downarrow$) | Corr. ($\uparrow$) | Dev. ($\downarrow$) | Corr. ($\uparrow$) | Dev. ($\downarrow$) | Corr. ($\uparrow$) |
| **Seed 0: Indep Set (IS)** | 32.66% | 0.26 | 25.29% | 0.47 | 25.41% | 0.47 |
| **Seed 1: Comb. Auction (CA)** | 30.01% | 0.34 | 21.40% | 0.53 | 23.44% | 0.55 |
| **Seed 2: Multiple Knapsack (KS)** | 33.84 % | 0.09 | 24.22% | 0.49 | 23.60% | 0.53 |
| **Seed 3: Generalized Indep. Set (GIS)** | 31.74% | 0.19 | 22.64 % | 0.51 | 25.61% | 0.49 |
| **Seed 4: Multi-Comm. Network Flow (NF)** | 36.49 % | 0.22 | 26.10% | 0.41 | 24.08% | 0.52 |
| **Seed 5: Set Cover (SC)** | 34.45 % | 0.11 | 24.80% | 0.45 | 26.13% | 0.47 |
| **Seed 6: Max Satisfiability (SAT)** | 43.07 % | 0.20 | 23.52% | 0.49 | 23.79% | 0.52 |
| **Seed 7: Cap. Fac. Location (CF)** | 33.00% | 0.09 | 20.39% | 0.58 | 25.24% | 0.51 |
| **Ours (Full Dataset)** | 20.14% | 0.58 | 20.14% | 0.58 | 21.56% | 0.59 |

### A.3.5 MIXING DIFFERENT FRACTIONS OF SEED V.S. *MILP-Evolve* GENERATED DATA.

We study the effect of mixing different fractions of Seed v.s. *MILP-Evolve* generated MILP instances on the performance for the Integrality Gap Prediction task. Specifically, we construct different training sets by varying the ratio of seed and instances generated from *MILP-Evolve* classes. We fix the total number of training and validation instances as 1200 and 300 instances, respectively. We then train a model on each of these training sets and test on the *MILP-Evolve* hold out test set (Table 1). From the result table below, we see that **including more MILP instances from *MILP-Evolve* (i.e. more diverse MILP classes) improve the performance.**

Table 13: Integrality Gap Prediction: Performance when mixing different fractions of Seed v.s. *MILP-Evolve* generated MILP instances as training and validation set, fixing the total number of training and validation instance constant. We test on the *MILP-Evolve* held out test set in Table 1.

|  | Seed 100% | Seed 80% + Evolve 20% | Seed 60% + Evolve 40% | Seed 40% + Evolve 60% | Seed 20% + Evolve 80% |
|---|---|---|---|---|---|
| Deviation (↓) | 32.96 | 25.66 | 23.57 | 21.67 | 21.32 |
| Correlation (↑) | 0.10 | 0.41 | 0.49 | 0.55 | 0.57 |

### A.3.6   PRACTICAL APPLICATION OF LANGUAGE-MILP CONTRASTIVE LEARNING TASK

We answer this question on two fronts: the utility of this task from the perspective of understanding MILPs and the potential of contrastive learning technique itself.

Many open source MILP datasets such as MIPLIB and in many business scenarios, the MILP instance files contain only constraints and variables (the raw $A, b, c$ values in the optimization), which are typically hard to understand and massive in size. Most of these MILP files lack descriptions of the underlying optimization problem and/or not sufficiently meaningful. Moreover, we cannot directly feed them into LLMs to interpret and generate language descriptions of the MILP formulations due to the context length limit, as seen from Table 14 and discussed in the next section.

Hence, this work takes first step with a contrastive learning approach to align GNN embedding of MILP instances with the text embeddings, aiming to provide meaningful interpretations when giving the MILP instances as input. Our results indicate that our approach holds lot of promise.

Given the abstract nature of the MILP instances, we believe any assistance in helping users' understanding of them is crucial. This can help nonexperts to understand the problem and also identify the incorrect formulations. This task also complements our other two tasks which are concerned with solving MILPs rather than understanding. We believe that a foundation model for MILPs that aims to democratize solving MILPs should also have the ability to help users to understand them.

### A.3.7   GPT ONLY BASELINE FOR LANGUAGE-MILP CONTRASTIVE LEARNING.

Table 14: Language-MILP Contrastive Learning: Performance Comparison of GPT-4o on the (sub-sampled) MILP instance files and the GNN-LLM contrastive model results.

|  | GPT-4o | Train From Scratch | Seed | Seed + Param Search | Seed + VAE | Ours |
|---|---|---|---|---|---|---|
| 4-Way Acc. (↑) | 47.79% | 72.37% | 72.20% | 75.17% | 72.90% | **77.62%** |
| 10-Way Acc (↑) | 16.81% | 46.50% | 42.45% | 42.66% | 44.61% | **53.99%** |

For the Language-MILP Contrastive Learning task, an interesting question is, whether we can use large language models (LLMs) to directly interpret the MILP instance to select the language alignment, hence bypassing the need to embed the MILP instance with the Graph Neural Network (GNN).

One of the bottlenecks with LLM directly interpreting MILP instances is that, the MILP instance files are typically huge as they contain the raw numerical values of the variables and constraints, substantially surpassing the context length of LLMs – that's why in this work, we focus on using GNNs to embed MILP instances, which can handle large MILP instances, to contrastive with the text embeddings from a language model.

We perform the experiment by subsampling the rows of the instance files up to context length. Specifically, we preprocess the MPS content for GPT-4o by anonymizing identifiers (e.g., problem names, variable names) and selecting 150 representative lines: 50 from the header, 50 randomly chosen from the middle, and 50 from the tail. This approach ensures sufficient context about the MPS file while staying within content length limits. We then construct a multiple-choice question and prompt GPT-4o to provide a step-by-step chain-of-thought analysis before making an educated guess.

Table 14 presents the results on the same *MILP-Evolve* test set as in Table 2. We observe the significant performance gap between GPT-4o and the GNN-LLM alignment methods as in the paper. This shows it is insufficient to use LLMs directly on the MILP-Language Contrastive Learning task.

An example GPT-4o's prompt and answer can be found next.

---

### An example prompt for GPT-4o to Perform the MILP-Language Contrastive Learning Task

You are provided with an MPS file representing a mixed integer programming (MIP) problem. Based on the content of this file, determine which of the following problems it is most likely associated with.

#### MPS FILE CONTENT

```
* SCIP STATISTICS
*   Variables: 3440 (3328 binary, 0 integer, 0 implicit integer, 112 continuous)
*   Constraints: 6320
OBJSENSE
   MIN
ROWS
 N  Obj
 L  Z_0
 L  Z_1
 L  Z_2
 ... 150 lines in total here. Skipped in LaTeX for conciseness ...
 L  C_96
 L  C_94
 L  C_102
 L  C_105
    x_5                C_94                               1
C_73                                    -1
    x_13               C_83                               1
C_89                                    1
 ...
ENDATA
```

#### CHOICES

**A:** This optimization model addresses a job-shop scheduling problem that incorporates machine and resource constraints, as well as precedence and priority requirements among jobs. Each job has a randomly assigned processing time, machine assignment, and resource requirement. The model also enforces group-based precedence constraints, dictating the order in which certain groups of jobs must be completed. Additionally, jobs assigned to the same machine are constrained by sequencing requirements to avoid overlap. A key component of the model is the inclusion of machine-specific capacity limits, ensuring that the cumulative resource requirements for jobs on a machine do not exceed its capacity. The objective function seeks to minimize the makespan, or the total time to complete all jobs, while also factoring in job affinities (reflecting job priority) and resource utilization. This combined objective is designed to promote efficient machine usage, balance workload, and respect priority allocations in high-demand scheduling environments.

**B:** The max cut optimization problem involves dividing the nodes of a graph into two groups to maximize the sum of weights on edges that have one endpoint in each group. Imagine you have a network where nodes are connected by weighted edges, and you want to split the nodes into two sets so that the "cut" between them (the edges connecting nodes in different sets) has the highest possible total weight. In this code, a graph is generated with random weights on edges, and a binary variable is assigned to each node to indicate its group. Constraints are added to ensure each edge either connects nodes in the same set or across sets. The objective function then maximizes the weight of edges across the cut, solving the problem using optimization techniques. This approach is useful in network design, where maximizing the separation or load between two groups is desired.

**C:** This multi-item lot-sizing problem models a scenario where a company needs to plan production and inventory for multiple products over a set number of time periods to minimize costs. For each period, the company faces specific demands for each product and must decide whether to produce it, how much to produce, and how much inventory to hold over time. Each production decision has an associated setup cost (for preparing machines or equipment), a holding cost (for storing any leftover inventory), and limits on how much can be produced in a period due to resource constraints. The model ensures that all customer demand is met while keeping costs low by balancing production quantities, setup times, and inventory levels across periods. This approach is widely used in manufacturing and supply chain management to optimize production schedules, reduce costs, and efficiently manage resources over time.

**D:** This optimization model is designed to identify optimal locations for electric vehicle (EV) charging stations within a given network of potential nodes. The model operates on a graph structure representing the network, where each node signifies a potential station location and each edge denotes connectivity between locations. It considers various factors, such as land costs, energy availability, zoning compatibility, grid substations, and tariff impacts. Each node has binary decision variables indicating station placement, resource allocation, and compliance with zoning regulations. The objective function seeks to maximize connectivity and substation resources while minimizing land costs and tariff impacts. Additional constraints ensure that energy demands do not exceed availability, and a priority variable with a large constant (Big M) term enforces priority constraints for specific nodes. The model's outcome identifies a balanced solution for station deployment that maximizes network accessibility and minimizes operational costs, while meeting regulatory and resource limitations.

#### INSTRUCTIONS

Think through the structure, variables, and constraints provided in the MPS file. Evaluate their alignment with the listed problem types. Justify your reasoning step by step before selecting the final answer.

Finally, provide only the single letter (A–D) that corresponds to your chosen answer in the format below. The last letter in your response will be used for evaluation.

**FINAL_ANSWER**: [A/B/C/D]

---

---

**An example GPT-4o Response to the MILP-Language Contrastive Learning Task**

## ANALYSIS OF THE MPS FILE

To identify which problem the provided MPS file is most likely associated with, let's analyze the structure and characteristics listed in the file and match them with the given choices.

### 1. VARIABLES

- The MPS file contains a total of 3440 variables, with 3328 binary variables and 112 continuous variables.
- Binary variables typically indicate decisions of whether or not to engage in specific actions (e.g., opening facilities, assigning specific nodes to a group, etc.).

### 2. CONSTRAINTS

- There are 6320 constraints present in the problem, a significant number which suggests complex interactions, common with problems involving combinatorial assignment and network design.

### 3. OBJECTIVE

- The objective is one of minimization (`OBJSENSE MIN`), which often implies cost minimization or minimizing some resource usage.

## MATCHING WITH CHOICES

Given these observations, let's match the structure to the choices:

### OPTION A (JOB-SHOP SCHEDULING)

This problem typically involves assignment and sequencing constraints, requiring integer and binary variables for ordering and timing decisions. While binary variables are used, it may not fit the minimization objective or the large number of binary variables if detailed resource constraints, as described, are absent.

### OPTION B (MAX CUT PROBLEM)

This involves dividing nodes into two groups, which aligns well with binary variable definitions. However, the problem typically emphasizes maximization of cut weights, not minimization.

### OPTION C (LOT-SIZING PROBLEM)

This problem focuses on production planning, inventory levels, and setup costs. While it involves minimization, the variable structure typically includes more continuous and integer variables, and it is less binary-heavy compared to the structure of the given MPS file.

### OPTION D (OPTIMAL LOCATION FOR EV CHARGING STATIONS)

This problem involves deciding locations, which aligns well with a binary variable for placement decisions. The large number of constraints fits with logistical, regulatory, and resource allocation concerns. Additionally, the focus on minimizing land costs matches the minimization objective.

## CONCLUSION

Taking into consideration the large number of binary variables, minimization objective, and extensive constraints, **Option D** seems to be the best fit. This type of problem typically has numerous binary variables for potential location decisions and many constraints related to logistical, regulatory, and resource allocation concerns, aligning well with the characteristics of the given MPS file.

## FINAL ANSWER

**FINAL_ANSWER**: D

**Example Description from the Test Set**

This optimization model addresses a complex job-shop scheduling problem with various constraints and objectives aimed at minimizing overall operational costs and delays. Each job has a processing time, tariff impact (costs affected by international tariffs), setup time between jobs on the same machine, and energy consumption rate. The model incorporates precedence constraints to ensure certain jobs are completed before others and assigns jobs to specific machines while managing limited machine capacity. Additional complexity is introduced through a machine breakdown risk managed by auxiliary variables, with penalties applied if breakdowns occur. The objective function minimizes the makespan (completion time of the last job), risk and energy costs, setup times between jobs, and potential breakdown penalties. This model enables efficient scheduling by balancing energy consumption, setup times, and breakdown management, ultimately supporting cost-effective job sequencing and machine utilization.

**Another Example Description from the Test Set**

This model focuses on optimizing the placement and phased deployment of electric vehicle (EV) charging stations across a graph-represented network. Each potential station site (node) is evaluated based on land costs, energy availability, zoning compatibility, and tariff impacts, with connections between nodes representing possible network benefits. Placement decisions involve constraints on energy resources and land use compatibility. The model includes deployment timing variables to enforce a phased installation sequence, ensuring stations are deployed in a specified order. The objective function seeks to maximize network accessibility—enhancing connections while considering node-specific weights—while minimizing associated land costs and tariff impacts. By accounting for setup costs, regional resource limits, and sequential deployment, the model provides a strategy for an efficient, cost-effective expansion of EV charging infrastructure.

**Example Description of the Seed Problem**

The provided MPS file outlines a mathematical model for a combinatorial auction problem, structured for a mixed integer programming formulation. The model aims to maximize the total price of accepted bids, employing binary decision variables to denote whether a bid is accepted. Each item is constrained to be included in at most one accepted bid, with inequality constraints set on the items ensuring this limit. The objective function involves a maximization with coefficients provided for the bids, while RHS values impose constraints such that certain items may appear at most once. This structured information allows one to understand how the model strategically enforces constraints and aims to achieve the highest possible total price for the accepted bids while ensuring no item overlaps. Additionally, performance details and model-specific techniques are noted to highlight optimization and study aspects within the combinatorial auction domain.

**Example Description for MILPLIB Dataset**

A problem in wireless networks. The objective is to select a minimum number of relay nodes so that any two nonadjacent nodes can communicate by way of the chosen relay nodes in at most $s$ hops, where $s$ is a problem input. The 2-hop case of this problem can be formulated as a set cover/hitting set problem with $n$ binary variables and $n^2$ constraints:

$$\sum_{k \in N(i) \cap N(j)} x_k \geq 1 \quad \text{for nonadjacent node pairs } \{i, j\}.$$

Despite the formulation's simplicity, instances with as few as 120 variables are left unsolved after one hour using Gurobi 7.0.2.

### A.4 PROMPT DETAILS

### A.4.1 MILP CODE SYNTAX

```python
import random
import time
import numpy as np
from pyscipopt import Model, quicksum

class ConferenceRoomScheduling:
    def __init__(self, parameters, seed=None):
        for key, value in parameters.items():
            setattr(self, key, value)

        self.seed = seed
        if self.seed:
            random.seed(seed)
            np.random.seed(seed)

    ################# Data Generation #################
    def generate_instance(self):
        assert self.min_capacity >= 0 and self.max_capacity >= self.min_capacity

        # Generate room capacities
        room_capacities = self.min_capacity + (self.max_capacity - self.min_capacity)
                        * np.random.rand(self.number_of_rooms)

        meetings = []

        # Create meeting schedules
        for _ in range(self.number_of_meetings):
            required_capacity = self.min_capacity + (self.max_capacity -
                                    self.min_capacity) * np.random.rand()
            start_time = random.randint(0, self.max_time - self.meeting_duration)
            end_time = start_time + self.meeting_duration

            meetings.append((required_capacity, start_time, end_time))

        room_availability = [[] for room in range(self.number_of_rooms)]
        for i, (required_capacity, start_time, end_time) in enumerate(meetings):
            for room in range(self.number_of_rooms):
                if room_capacities[room] >= required_capacity:
                    room_availability[room].append(i)

        ### given instance data code ends here
        ### new instance data code ends here

        return {
            "meetings": meetings,
            "room_availability": room_availability,
            "room_capacities": room_capacities
        }

    ################# Optimization Modeling (PySCIPOpt) #################
    def solve(self, instance):
        meetings = instance['meetings']
        room_availability = instance['room_availability']

        model = Model("ConferenceRoomScheduling")

        # Decision variables
        schedule_vars = {(r, i): model.addVar(vtype="B", name=f"Room_{r}_Meeting_{i}")
                        for r in range(self.number_of_rooms) for i in range(len(meetings))}

        # Objective: maximize the number of meetings scheduled
        objective_expr = quicksum(schedule_vars[r, i] for r in range(self.number_of_rooms)
                                for i in range(len(meetings)) if i in room_availability[r])

        # Constraints: Each room can host only one meeting at a given time
        for r in range(self.number_of_rooms):
            for i1 in range(len(meetings)):
                if i1 not in room_availability[r]:
                    continue
                for i2 in range(i1 + 1, len(meetings)):
                    if i2 not in room_availability[r]:
                        continue
                    if meetings[i1][1] < meetings[i2][2] and meetings[i2][1] < meetings[i1][2]:
                        model.addCons(schedule_vars[r, i1] + schedule_vars[r, i2] <= 1,
```

```
77                                        f"Room_{r}_Conflict_{i1}_{i2}")
78
79          # Constraints: Each meeting should be scheduled in at most one room
80          for i in range(len(meetings)):
81              model.addCons(quicksum(schedule_vars[r, i] for r in range(self.number_of_rooms)
82                                     if i in room_availability[r]) <= 1, f"Meeting_{i}")
83
84          model.setObjective(objective_expr, "maximize")
85
86          ### given constraints and variables and objective code ends here
87          ### new constraints and variables and objective code ends here
88
89          start_time = time.time()
90          model.optimize()
91          end_time = time.time()
92
93          return model.getStatus(), end_time - start_time
94
95  if __name__ == '__main__':
96      seed = 42
97
98      ################# Parameters #################
99      parameters = {
100          'number_of_rooms': 30,
101          'number_of_meetings': 125,
102          'min_capacity': 30,
103          'max_capacity': 300,
104          'meeting_duration': 20,
105          'max_time': 72,
106      }
107      ### given parameter code ends here
108      ### new parameter code ends here
109
110      scheduler = ConferenceRoomScheduling(parameters, seed)
111      instance = scheduler.generate_instance()
112      solve_status, solve_time = scheduler.solve(instance)
113
114
115      print(f"Solve_Status:_{solve_status}")
116      print(f"Solve_Time:_{solve_time:.2f}_seconds")
```

A.4.2  EXAMPLE PROMPTS

---

**Formulation Add Prompt**

Follow these step-by-step instructions to generate a diverse and realistic MILP by adding complexity to a given MILP using a specific MILP formulation method:

1.  Describe the Given MILP in Detail:
    - Place the MILP within a specific real-world context.
    - Clearly outline the context, constraints, objectives, and variables, emphasizing its practical significance.

2.  Choose a MILP Formulation Method from the following options: {three_random_formulation_methods}
    - Explain how this method applies to the given MILP in a real-world context.
    - Discuss the relevance of the chosen method to enhancing the complexity and realism of the MILP formulation.

3.  Summarize and Formulate a New MILP:
    - Retain the original MILP's structure but add complexity by incorporating the specific MILP formulation method.
    - Discuss how to introduce new constraints, variables, data, or change the objective function in the context of the real-world application.
    - Ensure new constraints or objectives are linear. Use tricks to linearize if necessary.

4.  Generate New MILP Code Step-by-Step as follows:
    - [Add Requirements, see Appendix A.4.3] *(We set* `prompt_optim="reflect the given MILP formulation method"` *for formulation add prompt.)*
    - [General Requirements, see Appendix A.4.4]

5.  Output Format:
    - Description of the given MILP:
      ⟨text to explain of the given MILP and its real-world application ⟩
    - Description of the chosen MILP formulation method and its application to the given MILP:
      ⟨text to explain the chosen MILP formulation method, its relevance to the given MILP in the context of the real-world application ⟩
    - Description of New Constraints, Variables, Objectives, Data, or Parameters:
      ⟨text to describe how to incorporate the specific formulation method to the given MILP by proposing new constraints, variables, objectives, data, or parameters⟩
      ⟨New [constraint/variable/objective/data/parameter] 1: description,
        New [constraint/variable/objective/data/parameter] 2: description,
        ...⟩
    - New complete MILP code:
      '''python
      ⟨complete code that incorporates modified constraints, data generation, and parameters⟩
      '''

6.  Here is the given MILP Code: {code}

---

**Cross-Over Prompt**

Follow these step-by-step instructions to generate a diverse and realistic MILP by incorporating information from another MILP to add complexity to a given MILP:

1. Provide a Detailed Description of the given MILP code:
   - Embed the MILP within a specific real-world application.
   - Clearly describe the context, constraints, objective, and variables in a way that highlights its practical importance.

2. Provide a Detailed Description of the second MILP code that you should incorporate into the given MILP code:
   - Embed the MILP within the same specific real-world application as the given MILP code.
   - Clearly describe the context, constraints, objective, and variables in a way that highlights its practical importance.
   - Explain the similarities and differences between the given MILP and the second MILP.

3. Explain the Modifications to the given MILP code and Their Impact:
   - Discuss how incorporating constraints, variables, or changing the objective function from the second MILP code to the given MILP code in the given real-world scenario.
   - The new MILP should retain the majority of the given MILP's structure, but make significant addition based on the second MILP codebase to make it more complex and challenging.
   - Ensure new constraints or objectives are linear. Use tricks to linearize if necessary.

4. Generate New MILP Code Step-by-Step as follows
   - [Add Requirements, see Appendix A.4.3] *(We set* `prompt_optim="reflect a combination of both MILP code"` *for cross-over prompt.)*
   - [General Requirements, see Appendix A.4.4]

5. Output Format:
   - Description of the given MILP:
     ⟨text to explain of the given MILP and its real-world application ⟩
   - Description of the second MILP:
     ⟨text to explain of the second MILP in terms of the similarities and differences between the given MILP and the second MILP in a real-world application context ⟩
   - Description of New Constraints, Variables, Objectives, Data, or Parameters:
     ⟨text to describe how to incorporate the second MILP to the given MILP by proposing new constraints, variables, objectives, data, or parameters ⟩
     ⟨New [constraint/variable/objective/data/parameter] 1: description,
       New [constraint/variable/objective/data/parameter] 2: description,
       ...⟩
   - New complete MILP code:
     '''python
     ⟨complete code that incorporates modified constraints, data generation, and parameters⟩
     '''

6. Here is the given MILP Code: {code}
   Here is the second MILP code: {code2}

---

**Mutate Prompt**

Follow these step-by-step instructions to generate a diverse and realistic MILP by slightly modifying the given MILP code:

1. Provide a Detailed Description of the given MILP code:
   - Embed the MILP within a specific real-world application.
   - Clearly describe the context, constraints, objective, and variables in a way that highlights its practical importance.

2. Provide a Detailed Description of new MILP code to suite a different real world application.
   - Discuss the novelty of the modifying constraints, variables, or changing the objective function from the given MILP by embedding it in a different real-world scenario.
   - Explain the relevance of these changes, detailing the similarity and differences between the original and new MILP.
   - The names of the new constraints, variables, or data should be full words starting with one of the letter {five_random_letters}.

3. Generate the new MILP code by following these procedures step-by-step:
   - [Mutate Requirements, see Appendix A.4.3] *(We set* `prompt_optim="reflect the new real world application"` *for the general mutate prompt.)*
   - [General Requirements, see Appendix A.4.4]

4. Output Format:
   - Description of the given MILP:
     ⟨text to explain of the given MILP and its real-world application ⟩
   - Description of the new MILP:
     Names: ⟨The full words of the names of constraint/variable/objective/data/parameter which starts with the capital letter selected from above. ⟩
     ⟨text to explain the new MILP and its real-world application ⟩
   - New complete MILP code:
     ```python
     ⟨complete code that incorporates modified constraints, data generation, and parameters⟩
     ```

5. Here is the given MILP Code: {code}

---

**Topic New Prompt**

Follow these step-by-step instructions to generate a diverse and realistic MILP following the given MILP code structure:

1. Provide a Detailed Description of the python coding style of the given MILP code.

   - The description should only focus on the code style, but not the MILP details.

2. Provide a Detailed Description of new MILP code to suite a different optimization problem under the topic {topic} with a specific real world application.

   - Discuss the novelty of the new MILP code in terms of constraints, variables, objective function, data and parameters and how they align with the given topic and the associated real-world scenario.
   - The names of the new constraints, variables, or data should be full words starting with one of the letter {five_random_letters}.

3. Generate the new MILP code by following the coding style of the given MILP code. For example, it must contain the following components

   - [New Requirements, see Appendix A.4.3] *(We set* `prompt_optim="reflect the given topic and the associated real world application"` *for topic new prompt.)*
   - [General Requirements, see Appendix A.4.4]

4. Output Format:

   - Description of the python coding style of the given MILP:
     ⟨text to explain of the python coding style ⟩
   - Description of the new MILP:
     Names: ⟨The full words of the names of constraint/variable/objective/data/parameter which starts with the capital letter selected from above.⟩
     ⟨text to describe how to incorporate the specific topic to generate the new MILP ⟩
   - New complete MILP code:
     ```python
     ⟨complete code that incorporates modified constraints, data generation, and parameters ⟩```

5. Here is the given MILP Code: {code}

---

---

**Delete Prompt**

Follow these step-by-step instructions to generate a diverse and realistic MILP by removing less important components from the given MILP code:

1. Provide a Detailed Description of the given MILP code.

2. Provide a Detailed Description of the new MILP code that removes the less important and components.

   • Explain how the new MILP removes the less important components from the given MILP.

   • You may modify the constraints, variables, or objectives, data generation or parameters to make the resulting MILP coherent.

   • The resulting MILP should still be complex and challenging to solve.

3. Generate the new MILP code by following these procedures step-by-step:

   • [Delete Requirements, see Appendix A.4.3]

   • [General Requirements, see Appendix A.4.4]

4. Output Format:

   • Description of the given MILP: ⟨text to describe in the given MILP ⟩

   • Description of the new MILP: ⟨text to explain what need to be changed to remove the less important components from the given MILP ⟩

   • New complete MILP code:
     ```python
     ⟨complete code that incorporates modified constraints, data generation, and parameters⟩
     ```

5. Here is the given MILP code {code}

---

### A.4.3 PROMPT-TYPE SPECIFIC REQUIREMENTS

---

**Add / Cross-Over Requirements**

- Add Constraints, Variables, or Objectives:
  - Introduce new elements to increase problem complexity while ensuring feasibility.
  - Place these in the `solve` function between ### given constraints and variables and objective code ends here and ### new constraints and variables and objective code ends here to {prompt_optim}.
- Modify Data Generation:
  - Use functions like `random.rand` or `random.randint` or `np.random.normal` or `np.random.gamma` or `nx.erdos_renyi_graph` or `nx.barabasi_albert_graph` or similar functions to create diverse datasets.
  - Insert new data in the `get_instance` function between ### given instance data code ends here and ### new instance data code ends here to support the added constraints, variables, or objectives.
- Update the Parameters:
  - Define new parameters in `if __name__ == '__main__'` between ### given parameter code ends here and ### new parameter code ends here to support the data generation for the optimization.
  - The value of each parameter should be a constant (e.g. integer, float, boolean, string). That is, if there is a tuple value, you should break down the tuple into individual parameters with constant values. If it is a more complicated data structure (a list, dictionary, set, or a function), please put the data structure in the `get_instance` function and only put the required constants to construct the data structure in the parameters dictionary.
  - Ensure parameters are adjustable to scale the problem's complexity.

---

**Mutate Requirements**

- Modify Realistic Constraints, Variables, or Objectives:
  - Modify the 'solve' function by updating the constraints, variables, or objective to {prompt_optim}.
  - Ensure the modifications challenge the solver and vary in difficulty, contributing to diverse solving times and gap improvements.
- Modify the Data Generation Procedure:
  - Modify the `get_instance` function by updating the res dictionary to support the modified constraints, variables, or objectives.
  - Use functions like `random.rand` or `random.randint` or `np.random.normal` or `np.random.gamma` or `nx.erdos_renyi_graph` or `nx.barabasi_albert_graph` or similar functions to create datasets of varying sizes.
- Modify the Parameters:
  - Modify the parameters within the `if __name__ == '__main__'` block to support the data generation for the optimization.
  - The value of each parameter should be a constant (e.g. integer, float, boolean, string). That is, if there is a tuple value, you should break down the tuple into individual parameters with constant values. If it is a more complicated data structure (a list, dictionary, set, or a function), please put the data structure in the `get_instance` function and only put the required constants to construct the data structure in the parameters dictionary.
  - Ensure parameters can be modified easily to scale up the problem or to alter its complexity.

---

**New Requirements**

- Generate Realistic Constraints, Variables, or Objectives:
  - Inside the `solve` function, define the constraints, variables, or objective to {prompt_optim}.
  - Ensure the new optimization problem challenge the solver and vary in difficulty, contributing to diverse solving times and gap improvements.
- Generate the Data Generation Procedure:
  - Inside the `get_instance` function, define the data generation and the res dictionary to support the new constraints, variables or objective.
  - Use functions like `random.rand` or `random.randint` or `np.random.normal` or `np.random.gamma` or `nx.erdos_renyi_graph` or `nx.barabasi_albert_graph` or similar functions to create datasets of varying sizes.
  - Ensure the new generated data support the generated constraints, variables, or objectives.
- Generate the Parameters:
  - Under the `if __name__ == '__main__'` block, generate the parameters to support the data generation for the optimization.
  - The value of each parameter should be a constant (e.g. integer, float, boolean, string). That is, if there is a tuple value, you should break down the tuple into individual parameters with constant values. If it is a more complicated data structure (a list, dictionary, set, or a function), please put the data structure in the `get_instance` function and only put the required constants to construct the data structure in the parameters dictionary.
  - Ensure parameters can be modified easily to scale up the problem or to alter its complexity.
- Avoid Repeating Existing Code. Make the modified constraints, variables, objectives, data, and parameters distinct from the original.

**Delete Requirements**

- Modify Realistic Constraints, Variables, or Objectives:
  - Modify the `solve` function by removing less important constraints or variables, or simplifying the objectives.
- Modify the Data Generation Procedure:
  - Modify the `get_instance` function by updating the res dictionary to support the modified optimization modeling.
  - Use functions like `random.rand` or `random.randint` or `np.random.normal` or `np.random.gamma` or `nx.erdos_renyi_graph` or `nx.barabasi_albert_graph` or similar functions to create datasets of varying sizes.
  - Ensure the modified data support the modified constraints, variables, or objectives.
- Modify the Parameters:
  - Modify the parameters within the `if __name__ == '__main__'` block to support the data generation.
  - The value of each parameter should be a constant (e.g. integer, float, boolean, string). That is, if there is a tuple value, you should break down the tuple into individual parameters with constant values. If it is a more complicated data structure (a list, dictionary, set, or a function), please put the data structure in the `get_instance` function and only put the required constants to construct the data structure in the parameters dictionary.
  - Ensure parameters can be modified easily to scale up the problem or to alter its complexity.

### A.4.4 GENERAL REQUIREMENTS

---

**General Requirements**

Ensure the Following:

- Code completeness:
  - Provide the COMPLETE, EXECUTABLE code, including all necessary library imports, the MILP class, and the parameters and code to call the MILP class.
  - Do not omit any part of the provided MILP code with '... (same as before) ...', even if it is not modified. Do not inherit from a previously defined class; instead, provide the entire codebase.
- Novelty and Increased difficulty for the new MILP while maintaining feasibility and reducing redundacy:
  - The new constraints, variables, objectives, data, and parameters types should be diverse, such as having different constraint types, and creative data generation schemes with correct syntax.
  - The complexity of the new MILP can be achieved by more complicated constraint, objective, data generation scheme or larger parameter values.
  - Avoid adding redundant constraints, variables, or objectives that do not contribute to the problem's complexity, but do provide a clear, concise, and complete executable MILP code.
- Modularity and executability of the new MILP code:
  - Maintain function integrity by ensuring that no references to out-of-scope variables are used.
  - Define any new helper functions within the `get_instance` or `solve` functions, ensuring they are correctly scoped and called.
  - Use clear, descriptive names for new parameters and variables, ensuring they align with the real-world context and adhere to correct syntax.
  - Ensure the code remains modular and can easily scale to larger or different MILP problems by simply adjusting parameters, without altering core functions.
- Do not include ### given constraints and variables and objective code ends here, ### new constraints and variables and objective code ends here, ### given instance data code ends here, ### new instance data code ends here, ### given parameter code ends here and ### new parameter code ends here in your final code.

---

