# OpenReview forum: "Towards Foundation Models for Mixed Integer Linear Programming"
_ICLR.cc/2025/Conference — ICLR 2025 Poster_

### Official Review · Reviewer_3Jap · 2024-10-30

**Soundness:** 3
**Presentation:** 3
**Contribution:** 3
**Rating:** 6
**Confidence:** 3

**Summary:**

This paper takes an early step to leverage foundation models for solving Mixed Integer Linear Programming (MILP), which plays an important role in real-world applications. Specifically, it studies three important tasks, including two typical tasks integrality gap prediction and learning to branch, and one proposed task of aligning MILP instances with natural language descriptions. Compared with previous works, it emphasizes generalization performance across problem classes, and proposes an LLM-based data augmentation framework named MILP-Evolve to generate diverse problem classes for the training of models. Experimental results demonstrate that the proposed MILP-Evolve can generate diverse data, and improve the overall performance of pretrained models on the three studied tasks.

**Strengths:**

1. Even that there have been a variety of works on leveraging LLMs to solve complex decision-making problems, to the best of my knowledge, this is the first work that focus on LLM-based training data augmentation in the field of learning to solve MILP. In experiments, the proposed MILP-Evolve show great capacity to generate diverse problems and improve the generalization performance of the trained models.
2. This paper is well-written, with rich technical details of the proposed MILP-Evolve. I am convinced that such a new open-source and powerful data argumentation method can benefit the community of learning to solve MILP.

**Weaknesses:**

1. As the idea of the proposed MILP-Evolve, which prompts the LLMs to generate diverse data under an evolution framework, is straightforward and not new, I am concerned that the technical insights of this paper are limited.

**Questions:**

1. What about the cost of data generation in the experiments? As the running of such a LLM-based data generation process may be very expensive, will the generated problem classes be collected and open-sourced together?

2. How do you envision the practical applications of the newly proposed task of aligning MILP instances with natural language descriptions in the MILP solving process? Can you discuss more on it?

---

> ### Author Response · Authors · 2024-11-21
> **Response to Reviewer 3Jap**
>
> **We are grateful for the reviewer's support of our paper and our commitment towards open sourcing the research to benefit the community. We provide our responses below, and kindly refer the reviewer to the general response for more experiments and discussions. We have also incorporated your feedback in the updated manuscript.**
>
>
> > As the idea of the proposed MILP-Evolve, which prompts the LLMs to generate diverse data under an evolution framework, is straightforward and not new, I am concerned that the technical insights of this paper are limited.
>
> Thanks for raising the concern. We kindly refer the reviewer to our **General Response 4 [GR4]** and our **Further Response to Reviewer fRvL (3)**, where we provide in-depth discussions of our contribution.
>
>
> > What about the cost of data generation in the experiments? As the running of such a LLM-based data generation process may be very expensive, will the generated problem classes be collected and open-sourced together?
>
> This is a great point. The cost of data generation and collection is indeed a major challenge towards training foundation models in general, and for MILPs in particular. That is why,  we will open source the generated problem classes so that other researchers with compute or budget restriction can easily use our data, and focus more improving the learning aspects.
>
> Having said that, our pipeline is also reasonably cost and compute efficient. To generate 100 valid classes costs less than $20 (GPT API costs) and take less than half a day (where the majority of the compute time is spent on parameter adjustment as it requires solving the instances from the generated classes). We note an interesting future direction is to potentially use neural surrogate or heuristic rules to speed up this parameter adjustment step.
>
>
> > How do you envision the practical applications of the newly proposed task of aligning MILP instances with natural language descriptions in the MILP solving process? Can you discuss more on it?
>
> We kindly refer the reviewer to our **General Response [GR2]** and our **Further Response to Reviewer fRvL (1)**, where we provide detailed explanations of the significance of the Language-MILP Contrastive Learning task.

---

> ### Author Response · Authors · 2024-12-01
> **Looking Forward to Your Reply**
>
> Dear Reviewer 3Jap,
>
> Thank you again for your valuable suggestions. We have conducted detailed experiments and provided thorough discussions to address your concerns. As the rebuttal period is nearing its conclusion, we wanted to follow up to ensure our responses have adequately addressed the reviewers' concerns. Please let us know if there are any remaining questions or areas where further clarification is needed. We sincerely appreciate the time and effort you have dedicated to reviewing our work. Thank you!
>
> Best,
>
> Authors

---

> > ### Comment · Reviewer_3Jap · 2024-12-02
> >
> > Thank you for the responses and kind reminder. I have also carefully reviewed the comments of other reviewers. Currently I have no more questions, and I will maintain my score as 6.

---

### Official Review · Reviewer_dtCu · 2024-11-03

**Soundness:** 3
**Presentation:** 4
**Contribution:** 2
**Rating:** 6
**Confidence:** 4

**Summary:**

This paper introduces MILP-Evolve, a novel LLM-based evolutionary framework designed to generate a large and diverse set of MILP instances. This paper evaluates the proposed method on three key learning tasks relevant to MILP: integrality gap prediction, learning to branch, and a new task of aligning MILP instances with natural language descriptions.

**Strengths:**

1. The proposed data generation method is validated on three MILP-related learning tasks.
2. The paper is well-structured and presented clearly.

**Weaknesses:**

1. The methodological contribution is somewhat limited, primarily offering a data augmentation approach that employs LLMs to generate diverse MILP instances.
2. There is a mismatch between the content and title of the paper. The title “Towards Foundation Models for Mixed Integer Linear Programming” suggests a broader scope, while the paper mainly discusses a data generation method for MILP.

**Questions:**

1. The fairness of experiments comparing the proposed data generation method with others is unclear. The paper mentions a 7:1:2 split of generated MILP problem classes into training, validation, and test subsets, raising concerns that the test data distribution may resemble that of the training data. In contrast, other methods likely produce differently distributed training data, which could skew comparisons and inflate the performance of MILP-Evolve.
2. For the task of aligning MILP instances with natural language descriptions, what are the specific formats for both the MILP instances and the textual descriptions? What are the sources of these instances and descriptions? Will all elements in each set be matched one-to-one, and does this task hold practical significance?
3. How should Figure 1b be interpreted?
4. The meaning and context of the “Mean” baseline used in the experiment is unclear.

---

> ### Author Response · Authors · 2024-11-21
> **Response to Reviewer dtCu**
>
> **We thank the reviewer for the constructive feedback and the time on effort spent on reviewing our paper. We provide our responses below, and kindly refer the reviewer to the general response for more experiments and discussions.**
>
> > The fairness of experiments comparing the proposed data generation method with others is unclear. The paper mentions a 7:1:2 split of generated MILP problem classes into training, validation, and test subsets, raising concerns that the test data distribution may resemble that of the training data.
>
> We thank the reviewer for raising this question. We have addressed your question in [GR1], where we have done additional experiments to test the generalization abilities of our model. We briefly summarize the discussion in [GR1] again here. We first refer the reviewer to our experiments on MIPLIB dataset in Section 5.3. We emphasize that our models are not pretrained on **MIPLIB dataset**, hence the results on MIPLIB are strong a indicator of the generalization abilities of our model. Moreover, during the rebuttal phase, we also performed additional experiments where we generated a new test dataset consisting of 50  new MILP classes generated using 6 seed classes that different from the ones used in training.
>
>
> > The methodological contribution is somewhat limited, primarily offering a data augmentation approach that employs LLMs to generate diverse MILP instances.
> >
> > There is a mismatch between the content and title of the paper. The title “Towards Foundation Models for Mixed Integer Linear Programming” suggests a broader scope, while the paper mainly discusses a data generation method for MILP.
>
> We kindly refer the reviewer to our General Response 4 [GR4], where we provide an in-depth discussion of our contribution and our choice of the title. We note that if the reviewer think this title can distract the audience from the main scientific contribution and have better suggestions, we would be happy to reconsider naming the paper differently in the future revisions.
>
>
> > For the task of aligning MILP instances with natural language descriptions, what are the specific formats for both the MILP instances and the textual descriptions? What are the sources of these instances and descriptions? Will all elements in each set be matched one-to-one, and does this task hold practical significance?
>
> We thank the reviewer for the question. Regarding the practical significance of the Language-MILP task, we refer the reviewer to our General Response 2 [GR2].
>
> The details of the instance and description generation can be found in Appendix A.1.5 and A.1.6. Specifically, the MILP instances and descriptions for the MILP-Evolve dataset are both obtained from MILP classes (the optimization code script). The MILP instances consist of $A, b, c$ matrices corresponding to linear constraints and objective function, which is same as the other two learning tasks. The language descriptions are generated based on both the MILP class code and extracted information from a rule-based parser. For the MIPLIB experiment, the MILP instances and descriptions are directly obtained from the MIPLIB webpage. Finally, for all the cases, MILP instance and language pairs are matched one-to-one.
>
> We hope this answers your question, and we have incorporated some of these details in our updated manuscript.
>
>
> > How should Figure 1b be interpreted?
>
> In Figure 1b, we take the code embedding of each MILP class (use OpenAI's text-embedding-ada), and perform TSNE to visualize them in the 2d space. The interpretation is that, originating from eight seed classes (the orange dots scattered around the space), the evolved classes gradually fill in the space, showing the diversity of the generated classes (at least at the code level).
>
>
> > The meaning and context of the “Mean” baseline used in the experiment is unclear.
>
> For the description of the mean baseline, we have modified the text in Sec. 5.1. to “For integrality gap prediction, we also include a Mean baseline, which, for all MILP instances, predicts the same constant value given by the mean of all the training set labels." That is, given all Integrality Gap labels in the training set $\\{y_i\\}_{D^{train}}$, we use the average $\frac{1}{|D^{train}|} \sum_i y_i$ as the prediction for all test instances. We hope the updated text clarifies the reviewer's question.

---

> > ### Comment · Reviewer_dtCu · 2024-11-26
> >
> > Thank you to the authors for the detailed responses. I have also reviewed the comments from the other reviewers and the authors' replies to all of them. It is clear that the authors have made significant efforts to improve the paper, but I still maintain the concerns that I initially raised:
> >
> > 1. **Limited Contribution:** I still believe the contribution of this paper is limited. Although the authors clarified their contribution by stating, "Our key motivation for the work is studying whether a foundation model approach -- that is, pretraining on a large and diverse data that can generalize to a wide variety of downstream tasks -- can be an effective paradigm for MILPs," I maintain my original view. Essentially, the contribution of this paper seems to be focused on a data generation method for MILP, and the innovation in methodology is limited. This is also echoed by Reviewer 3Jap, who mentioned: "the technical insights of this paper are limited."
> >
> > 2. **Questionable Contribution of Contrastive Learning Framework:** Regarding the contribution, in Section 1.1, the authors claim that one of their contributions is "A Contrastive Learning Framework for Understanding MILP in Natural Language." While the paper introduces this new task and proposes a corresponding solution, I feel that this contribution seems somewhat forced, as it does not directly relate to the core contribution of the paper, which is the MILP data generation method.
> >
> > 3. **Inappropriate Title for Contribution:** Based on the above two points, I still believe the title of the paper is inappropriate. "Towards Foundation Models for Mixed Integer Linear Programming" exaggerates the contribution of the paper and is too general, obscuring the specific contribution. A more appropriate title would be one that highlights their contribution to MILP data generation methods.
> >
> > 4. **Inappropriate Title for Trained Model:** Additionally, regarding the title, I understand the authors want to validate the effectiveness of their data generation method on multiple MILP tasks. However, the three MILP tasks studied in the paper are not well-integrated, despite the authors' clarifications. As Reviewer fRvL pointed out: “Disconnect Between Tasks.” Furthermore, the authors train independent models for each of these tasks, rather than a unified foundation model. While this approach is not inherently flawed, when considering the paper’s title, it gives the impression that the current work is still far from realizing the concept of a foundation model.
> >
> > Other concerns are as follows:
> >
> > 5. **Significance of Language-MILP Contrastive Learning Task:** Although the authors have added some clarification about the significance of the language-MILP contrastive learning task, I remain skeptical. Specifically, I find it hard to imagine a real-world scenario where there is a practical need to pair a set of MILP instances with a corresponding set of language elements in a one-to-one fashion if these two sets are pre-specified in advance.
> >
> > 6. **Insufficient Initial Submission:** Although the authors provided substantial clarifications during the rebuttal stage, I feel that the original submission had significant shortcomings. A good paper should ideally be relatively complete when first submitted (allowing for some minor issues or open questions), rather than relying heavily on important clarifications during the rebuttal phase.
> > ﻿
> > Considering these concerns, I maintain my score for the paper.

---

> > > ### Author Response · Authors · 2024-11-26
> > > **Further Response to Reviewer dtCu**
> > >
> > > We thank the reviewer for their reply.
> > >
> > > We want to point out that we have clarified in our rebuttal what we meant by "Towards a foundation model" — specifically, our methodology eliminates the need to train separate models for different MILP problem classes, addressing a significant research gap in the literature. However, we are more than happy to consider alternative titles, such as “*Efficient Multi-Class Learning for Mixed Integer Programming: An LLM-Based Data Augmentation Approach*”, to address the reviewer’s concern.
> > >
> > > Regarding the "disconnect between task,” it is noteworthy that the same data (MILP classes and instances) and GNN architecture backbones perform effectively across three different learning tasks. As outlined in our response and further emphasized at the end of Sec. 2.2 of the main paper, these tasks are complementary, focusing on understanding, predicting outcomes, and accelerating MILPs. Tackling these elements together is crucial for making MILPs more practically applicable.
> > >
> > > Finally, while the contrastive learning task may appear somewhat stylized, we view its study as an important step toward related tasks, such as directly generating descriptions from MILP instances. While we acknowledge that the generating human understandable descriptions from MILP is practically useful and represents the ultimate goal, we, along with the broader community, are not yet at that point. We strongly believe this work provides a meaningful first step toward that objective, similar to how CLIP contrastive learning paved the way for text to image generative model such as DALLE by training a decoder model on CLIP embeddings.
> > >
> > > Regarding the concerns about updating the submission during the rebuttal phase, we respectfully disagree with the framing of this as a negative practice. The conclusions and insights presented in our original submission remain unchanged, and the all the new experiments we did based on the reviewer's feedback corroborate the findings of the initial experiments. We view the ICLR rebuttal period as an opportunity to enhance and clarify the paper in response to feedback, fostering a collaborative and constructive review process. We hope this perspective resonates with the spirit of open and supportive discourse.

---

> > > > ### Comment · Reviewer_dtCu · 2024-11-26
> > > >
> > > > Thank you for the authors' further response.
> > > >
> > > > I believe the alternative title is more appropriate than the original one.
> > > >
> > > > I also do not think it is a negative thing to update a paper during the rebuttal period. This is just a small concern of mine, not my main one. I remain neutral and open-minded regarding whether the quality of the original submission should be assessed. On the contrary, I greatly appreciate the authors' significant efforts to improve the paper during the rebuttal period.
> > > >
> > > > Nevertheless, I still believe that the current version of the paper falls slightly below the ICLR threshold. Therefore, at this stage, I am maintaining my score. Meanwhile, I will wait for the opinions of the other reviewers.

---

> > > > > ### Author Response · Authors · 2024-12-02
> > > > > **Further Response to Reviewer dtCu**
> > > > >
> > > > > Dear Reviewer dtCu,
> > > > >
> > > > > Thank you again for the time and effort you put into reviewing our work.
> > > > >
> > > > > As the discussion period nears its conclusion, we would like to take this opportunity to inform you that, following our discussion with reviewer fRvL, we have provided a more detailed restatement of our contributions and further clarified the significance of the Language-MILP contrastive learning task. These discussions are reflected in the revised Contribution section (Sec. 1.1) and the Language-MILP task description section (Sec. 2.2.3).
> > > > >
> > > > > We bring this to your attention as we believe these discussions may further address your concerns. For more details, we kindly refer the reviewer to our **Further Response to Reviewer fRvL (1)** and **Further Response to Reviewer fRvL (3)**. We hope the reviewer find these responses satisfactory, and we are happy to answer any additional questions you may have.
> > > > >
> > > > >
> > > > > Thanks,
> > > > >
> > > > > Authors

---

> > > > > > ### Comment · Reviewer_dtCu · 2024-12-03
> > > > > >
> > > > > > Thank you for your efforts and clarification. I believe that the proposed data generation method does make a certain contribution to the MILP community, but my main concern is the limited novelty in methodology. I am not saying that this article lacks innovation, and I am just not sure whether such limited novelty can meet the threshold for ICLR. Given the authors' significant efforts in improving the quality of the paper, I am willing to increase my score to 6. Nonetheless, I remain neutral on whether this paper should be accepted at ICLR. Good luck.

---

> > > > > > > ### Author Response · Authors · 2024-12-03
> > > > > > > **Further Response to Reviewer dtCu**
> > > > > > >
> > > > > > > We appreciate the reviewer for acknowledging the value of our proposed MILP-Evolve data augmentation to the MILP community, and we thank the reviewer for increasing the score. We believe this paper introduces novel contributions that are highly valuable to the community in the intersection of learning and optimization:
> > > > > > >
> > > > > > > - By combining LLM-based data augmentation with GNN for MILP representation learning, our integrated learning framework allows us to *close a notable research gap by significantly improving the performance of GNN-based architectures in the multi-class learning setting*. This contrasts with prior works, which predominantly focus on training separate GNN models for each MILP class.
> > > > > > > - Our evolution-based data generation method, MILP-Evolve, has *demonstrated effectiveness across diverse learning tasks for understanding and solving MILPs*. By designing a framework that generates MILP instances with *desirable, controllable properties* such as feasibility and nontrivial solve time, we were able to create a first-of-a-kind MILP dataset with more than a thousand diverse MILP classes. The dataset and associated methodology hold *significant potential to enhance the generalizability of learning for many related tasks in the future*, such as improving MILP solvers in areas like presolving, scheduling primal heuristics, and generating cuts.
> > > > > > > - Our new Language-MILP contrastive learning task is an important stepping-stone in *systematically bridging the gap between recent NLP advancements and state-of-the-art learning models for MILP (GNNs)*. To the best of our knowledge, this is in contrast to previous studies that have predominantly focused on either GNNs or LLMs in isolation when studying MILPs.
> > > > > > >
> > > > > > > Our exploration of these novel directions has further provided *valuable insights (e.g. the importance of class diversity)* that we believe will significantly benefit the community. We are confident that this work is both exciting and capable of inspiring future research in the intersection of LLMs and optimization, and we are committed to fully open-sourcing our work, including our MILP-Evolve dataset, to support these future advancements in the community.

---

### Official Review · Reviewer_fRvL · 2024-11-04

**Soundness:** 2
**Presentation:** 2
**Contribution:** 2
**Rating:** 6
**Confidence:** 4

**Summary:**

The paper explores the potential of foundation models for Mixed Integer Linear Programming (MILP), introducing a novel framework called MILP-Evolve that leverages large language models (LLMs) to generate diverse MILP instances. The authors apply this framework to three distinct tasks: (1) integrality gap prediction, (2) learning to branch, and (3) a new task of aligning MILP instances with natural language descriptions. While promising empirical results are shown, especially in generalizing across different MILP classes, some aspects—particularly the Language-MILP Contrastive Learning task—require further clarification regarding its practical significance and feasibility.

**Strengths:**

- **Diversity of MILP Generation**: The MILP-Evolve framework introduces an innovative approach to generating diverse MILP problem classes, which has the potential to enhance generalization in ML-based MILP solvers.
- **Empirical Performance**: The paper demonstrates strong performance improvements on the integrality gap prediction and learning to branch tasks, providing evidence that the proposed approach can generalize to unseen MILP classes.
- **Extensive Experimental Work**: The paper presents a substantial amount of experimental results across various tasks, demonstrating the significant effort in evaluating the proposed methods. The authors cover a wide range of experiments, which showcases the robustness of their approach.

**Weaknesses:**

1. **Practical Application of Language-MILP Contrastive Learning**: The Language-MILP Contrastive Learning task is positioned as a way to assist non-experts in understanding and formulating MILPs. However, the generated natural language descriptions tend to emphasize technical details (e.g., linear constraints, integer variables), and it is not entirely clear how this helps users grasp the real-world significance of MILP problems. It would be helpful if the authors could provide more clarification on how aligning these mathematical descriptions with natural language assists in bridging the gap between abstract optimization models and their practical applications. Including more concrete examples or case studies could further reinforce this task’s practical relevance.

2. **Language Quality**: The natural language descriptions used in the Language-MILP Contrastive Learning task are generated by LLMs from solver code (e.g., SCIP, Pyomo, Gurobi). There may be some concerns regarding the quality of these descriptions. If the language samples were curated by human experts, this task could capture valuable domain-specific insights. However, relying solely on LLM-generated descriptions raises questions about the meaningfulness of the alignment. It might be worth considering how this task compares to having an LLM directly interpret a new MILP, as it is not immediately clear whether the proposed approach would outperform such a method. Clarifying the value added by this task would strengthen the paper.

3. **Disconnect Between Tasks**: While the paper introduces multiple tasks, the connection between them could be better articulated. For instance, the relationship between the Language-MILP Contrastive Learning task and the Multi-Class Learning task is not immediately clear, which might make the paper seem somewhat disjointed. A clearer explanation of how these tasks fit together within the broader scope of MILP optimization, particularly how Language-MILP Contrastive Learning complements the other optimization tasks, would improve the cohesion of the work.

4. **Comparative Experiments**: The comparison between the proposed method and works like ACM-MILP in the experiments might benefit from some adjustments. The current experimental setup involves:
   - Using problems generated by MILP-Evolve based on 8 seed MILP classes to create a large number of new problem types.
   - Using problems generated by ACM-MILP, which also learns and generates problems based on the same 8 seed MILP classes.

   Both sets of problems are then used as training data for a downstream ML-based MILP optimization framework, and the models are tested on other MILP classes generated by MILP-Evolve. However, existing MILP generation frameworks typically aim to enhance performance within a specific type of MILP class. Therefore, I suggest the following alternative comparative experiments:

   **Experiment 1:** Compare the models trained on problems generated by:
   - MILP-Evolve, which generates a large number of new problem types based on the 8 seed MILP classes.
   - ACM-MILP, which learns and generates problems based on the same 8 seed MILP classes.

   Then, test the trained MILP optimization frameworks on the same 8 seed MILP classes used by both MILP-Evolve and ACM-MILP. This would allow for a more direct comparison within the shared MILP classes.

   **Experiment 2:** Select a set of MILP problems generated by MILP-Evolve or from MIPLIB as seed MILP classes. Compare the models trained on problems generated by:
   - Problems generated by MILP-Evolve based on the selected seed MILP classes.
   - Problems generated by ACM-MILP based on the same selected seed MILP classes.

   After training, test both frameworks on the selected seed MILP classes to directly compare their performance.

   These alternative experimental designs would provide a more balanced comparison, as they ensure that both approaches have access to similar training data. This could help avoid potential biases in the current experimental setup, where ACM-MILP might be disadvantaged by the absence of instances from the test problem classes in its training set.

5. **Overclaim in Contribution**: The paper states that it achieves “Substantial Multi-Class Learning Gains Across All Tasks,” but the results presented primarily focus on integrality gap prediction and learning to branch. Since there are no substantial results or experiments demonstrating gains in the Language-MILP task, this claim could be seen as somewhat overstated. The authors could either provide additional results for the Language-MILP task or rephrase the contribution to more accurately reflect the scope of the work.

6. **Impact of Seed Class Selection**: The choice of seed classes in the MILP-Evolve framework likely has an important influence on the distribution and diversity of the generated MILP classes. However, the paper does not delve deeply into this aspect or provide an experimental evaluation of how seed class selection affects the generated instances. Including an analysis of how different seed classes influence the diversity and quality of the generated MILP instances, and whether certain seed classes lead to better generalization in the optimization tasks, would help strengthen the paper’s claims regarding the versatility of MILP-Evolve.

**Questions:**

1. Could the authors further clarify the real-world impact of the Language-MILP task? Specifically, how does aligning natural language descriptions with MILPs help non-experts understand and solve optimization problems?
2. Given that the language samples are generated by LLMs from solver code, how do the authors ensure the quality of these samples? Would human-generated descriptions lead to better learning outcomes in this task?
3. How does the Language-MILP Contrastive Learning task connect to the other tasks in the paper, such as integrality gap prediction and learning to branch? Could the authors provide more insights into the overall coherence of the tasks?

---

> ### Author Response · Authors · 2024-11-21
> **Response to Reviewer fRvL (1): New Experiments for Language-MILP Alignment**
>
> **We are grateful for the insightful and detailed feedback from the reviewer, which greatly strengthen our work. We provide our responses below, and kindly refer the reviewer to the general response for discussions and new experimental results.**
>
>
> > It might be worth considering how this task compares to having an **LLM directly interpret a new MILP**, as it is not immediately clear whether thproposed approach would outperform such a method. Clarifying the value added by this task would strengthen the paper.
>
> This is a great suggestion! We refer the reviewer to our general response section [GR2] where we did the experiment you asked for. Our result shows that GPT-4o performs poorly when directly interpreting the MILP instances.  One of the bottlenecks with LLM directly interpreting MILP instances is that, the MILP instance files are typically huge as they contain the raw numerical values of the variables and constraints, substantially surpassing the context length of LLMs; in [GR2], we subsample the rows of the instance files up to context length, but the missing information leads to subpar performance. That's why in this work, we focus on using Graph Neural Networks (GNNs) to embed MILP instances and perform contrastive learning with the text embedding of the description from a language model.
>
>
> > **Language Quality**. Given that the language samples are generated by LLMs from solver code, how do the authors ensure the quality of these samples? Would human-generated descriptions lead to better learning outcomes in this task?
>
> This is a nice question.  Inspired by your suggestion, in a new experiment we performed during rebuttal (see General Response 1 [GR1]), we checked the GPT generated descriptions and manually modified the descriptions to make sure the descriptions match the optimization problems. We observed that GPT-generated descriptions are generally accurate, correctly reflecting the optimization problem and highlighting potential real-world applications of the optimization class (e.g., "This type of scheduling is essential in manufacturing and project management, where minimizing total completion time across multiple tasks is critical."). Some descriptions included irrelevant details (e.g., ``PySCIPOpt is used to solve the optimization problem"), which we manually removed.
>
> Given these expert/human verified language labels, initializing with our model pre-trained on MILP-Evolve consistently outperform all the baselines, further strengthening contributions of our work. We appreciate your comment in pointing this out.
>
> Having said that, expert human annotations would indeed help in further improving the quality of model generated descriptions. Unfortunately, human expert labeling is time consuming and does not scale to large datasets.  Once we bootstrap a process towards training foundation models for MILP, one also has the potential to collect human annotations of the MILPs based on users feedback, similar to how AI math tutoring frameworks collect annotations from the users or in RLHF frameworks.  An interesting future direction is to explore more principled hybrid human and LLM labeling methods, and perform A/B testing to provide accurate and abundant description labels.

---

> ### Author Response · Authors · 2024-11-21
> **Response to Reviewer fRvL (2): Practical Application and Contribution of Language-MILP Alignment**
>
> > **Practical Application of Language-MILP Contrastive Learning.** Could the authors further clarify the real-world impact of the Language-MILP task? Specifically, how does aligning natural language descriptions with MILPs help non-experts understand and solve optimization problems.
>
> We kindly refer the reviewer to our General Response 2 [GR2], where we discuss the practical applicability of the Language-MILP Contrastive Learning Task. In summary, MILP instances that arise in large scale production systems are extremely large with constraint matrices spanning multiple files, and are hard to parse both for non-experts and LLMs directly. Using our technique, one can identify good descriptions of such MILP instances that are easy to understand.
>
> Here is one example of the MILP instance description: “This Loyalty Rewards optimization model is designed to maximize the total benefits from rewarding members within different regional capacity limits. Each member has a benefit weight, and each region has a resource limit. The model assigns rewards to members (represented by binary decision variables) such that the total rewards given in each region do not exceed its limit ..." We believe such descriptions can help non-experts understand the meaning of this optimization problem.
>
> More importantly, as MILP instances grow in size, without human readable descriptions, it may be impossible to find mistakes or improve them. Thus we foresee language-MILP alignment as a crucial component of a foundation model for MILPs. While our paper takes the first step towards this task and shows the technical feasibility of via a contrastive learning approach, we acknowledge that a lot needs to be done. Improving the quality of this alignment task is an important future research direction with immense practical value. As per your suggestion, we provide a discussion on the significance and importance of this task in our updated manuscript.
>
>
> > **Disconnect Between Tasks.** How does the Language-MILP Contrastive Learning task connect to the other tasks in the paper, such as integrality gap prediction and learning to branch? Could the authors provide more insights into the overall coherence of the tasks?
>
> The three tasks addressed in this work capture interconnected aspects of understanding and solving MILP instances: (1) the Language-MILP Alignment task aids the understanding of MILP instances' structure and characteristics. A deeper understanding of  MILP instances aids non-experts in comprehending the problem; it enables experts to develop, debug MILP instances, and design specialized algorithms based on instance-specific properties, (2) the Integrality Gap Prediction task focuses on analyzing solution properties of the MILP instance. An accurate integrality gap predictions can guide algorithm selection, allowing instances with tight gaps to be solved via LP relaxation without fully solving the MILP. (3) The learning to Branch task improves the process of solving MILP instances through more effective branching, which can have huge time and cost saving for industrial applications and production pipelines.
>
> Hence, these tasks are complementary and collectively essential for both practitioners and experts to advance MILP research.
>
>
> > **Overclaim in Contribution for Language-MILP.** Since there are no substantial results or experiments demonstrating gains in the Language-MILP task, this claim could be seen as somewhat overstated.
>
> Thank you for raising this concern.  We are worried that the reviewer may have a misunderstanding of our paper, and we would like to provide clarification here. All experiments we did on 3 test datasets show consistent and significant improvement on Language-MILP alignment task. This can be see from Table 1 in the main paper, the new experiment we did in the General Response 1 [GR1] section, and the MIPLIB results in the main paper. For example, in [GR1], initializing with **Ours** achieves a 10-Way accuracy of **53.99%**, whereas initializing with Seed + VAE (ACM-MILP) only has a 10-Way accuracy of 44.62%.

---

> ### Author Response · Authors · 2024-11-21
> **Response to Reviewer fRvL (3): Ablation Experiments - Additional Comparison with ACM-MILP & Impact of Seed Class**
>
> >  **Comparative Experiments with ACM-MILP.**
>
> We agree with the reviewer that “existing MILP generation frameworks (such that ACM-MILP) typically aim to enhance performance within a specific type of MILP class”. This pinpoints a research gap in the previous literature on learning to *generalize in the multi-class setting, which is exactly the focus of our paper* and greatly enabled by the MILP-Evolve generation procedure. As seen in the paper and the results on the new held out test set in our General Response [GR1], learning on classes generated by our MILP-Evolve pipeline significantly improves the generalization performance.
>
> We would like to point out that the difficulty of the learning tasks drastically increases in the heterogeneous multi-class settings in comparison to the homogeneous single class setting studied in ACM-MILP paper and in previous literature. To support this statement, in the following table, we conduct an experiment similar to that of  Experiment 1 suggested by the reviewer: we take the three models learned on (1) the seed classes (**Seed**), (2) augmented by ACM-MILP (**Seed+VAE**), and (3) our MILP-Evolve generated classes (**Ours**), and test on held out instances from the Seed classes -- that is, we consider instances within the seed classes (we use slightly different parameters from those in the training set to increase diversity in the test set). This process creates a test set that is in-distribution. On this test set, we see that all three models perform similarly, which makes sense as the three models have seen abundant in-distribution instances during training.
>
> Moreover, we find that learning on the seed classes here are easier than learning on the MILP-Evolve or MIPLIB test classes used in the paper.  For example, for integrality gap prediction, the deviation (around 10%) here is much lower than the deviation for the MILP-Evolve or MIPLIB test set (around 20%). This shows that learning within a few classes (as done in ACM-MILP) is much easier than learning in the multi-(MILP-)class setting (the focus of our paper).
>
> |                                         | Seed   | Seed + VAE | Ours   |
> | --------------------------------------- | ------ | ---------- | ------ |
> | Integrality Gap: Deviation ($\downarrow$) | 9.67%  | 11.86%     | 9.25%  |
> | Language-MILP: 4 Way Acc. ($\uparrow$)   | 52.18% | 53.71%     | 57.21% |
>
>
> >  **Impact of Seed Class Selection.** Including an analysis of how different seed classes influence the diversity and quality of the generated MILP instances, and whether certain seed classes lead to better generalization in the optimization tasks, would help strengthen the paper’s claims regarding the versatility of MILP-Evolve.
>
> We thank the reviewer for the great suggestion. We refer the reviewer to General Response 3 [GR3] for our detailed study on the effect of different seed classes.

---

> > ### Comment · Reviewer_fRvL · 2024-11-27
> >
> > Thank you to the authors for the detailed responses and additional experimental results. I greatly appreciate the effort put into the rebuttal, especially the new experiments related to the Language-MILP alignment task, seed class selection, and comparison with ACM-MILP. These additions address several concerns, but I still believe further clarification and refinement are needed to fully realize the potential of this work.
> >
> >  **Q1: Practical Applications of the Language-MILP Alignment Task:**
> > The authors state that the Language-MILP alignment task aids in understanding MILP instances, debugging issues, and designing specialized algorithms. However, the experiments focus primarily on aligning existing problems with pre-written descriptions, leaving the generalization ability to unseen MILPs unexplored. For practical applications, the ability to generate meaningful descriptions for new MILPs would be more impactful. Additionally, the claim that this task aids in debugging is not well-supported. For example, can the method detect or diagnose errors in constraints or objectives for large MILPs? Concrete examples or case studies demonstrating these capabilities would significantly enhance the contribution of this task.
> >
> > **Q2: Practical Implications of Integrality Gap Prediction:**
> > The authors argue that integrality gap prediction can guide algorithm selection, allowing tight-gap problems to be solved via LP relaxation, but its implementation remains unclear. Specifically, how does LP relaxation help solve MILPs with tight but non-zero gaps? Does it involve rounding heuristics or constraint adjustments? Furthermore, while the authors claim this task can reduce solve time or improve solution quality, these benefits are not explicitly demonstrated in the experiments. Providing real-world use cases or examples where integrality gap predictions improve solving efficiency would clarify its practical utility. Additionally, a discussion of robustness is necessary—how do prediction errors impact downstream tasks?
> >
> > **Q3: Contribution Scope and the Role of Foundation Models:**
> > While MILP-Evolve is a significant contribution for generating diverse MILP data, the broader claims about foundational models feel overstated. The work addresses three independent tasks (Language-MILP alignment, integrality gap prediction, and learning to branch), but no unified model or novel architecture is proposed. Instead, the primary innovation lies in data generation, as noted by another Reviewer dtCu . The subsequent tasks largely rely on existing methods, with no major innovations in their design. A more precise framing, such as emphasizing the data generation contribution, would align better with the paper’s actual scope. Additionally, MILP-Evolve’s performance is closely tied to seed class selection, and its generalizability to real-world problems remains uncertain. Experiments incorporating seed classes from real-world datasets like MIPLIB could help validate its broader applicability.
> >
> > Thank you again for your efforts and thoughtful responses.

---

> > > ### Author Response · Authors · 2024-11-27
> > > **Further Response to Reviewer fRvL (1)**
> > >
> > > **Thank you so much for your response. We are glad that our rebuttal addresses several of the reviewer’s concerns. We want to provide the following answers to your additional questions.**
> > >
> > > **Q1. Practical Applications of the Language-MILP Alignment Task.**
> > >
> > > We would like to clarify the importance of the alignment task as a way to assist non-experts and experts’ understanding of the MILP instances; we apologize for the unclear word choice of “debugging” that we used in the rebuttal. As stated in the updated introduction of the paper, this task complements the tractability tasks by lowering the entry barrier for non-experts by deepening their understanding of optimization formulations. We provide further clarification on the importance of this task below, and have revised Sec. 2.2.3 of the paper to include a summary of this discussion.
> > >
> > > We hope that the reviewer would agree that it would be extremely useful if there was a model that can generate human readable natural language descriptions of complex MILPs. How can one go about designing such a system? Here, we take inspiration from CLIP/DALLE model frameworks for generating images from textual descriptions. Our first insight is that we should treat text and MILP instances as different modalities similar to text and images.
> > >
> > > Now, let us understand the text-to-image (or image to text) generative models. They consists of two parts and 2 separate papers:
> > > 1. An embedding/encoder model that does contrastive learning to bring different modalities to a common space (CLIP paper [1])
> > > 2. Then, training a decoder model to invert these embeddings to a generative model (DALLE paper [2])
> > >
> > > We note that both these are hard technical challenges. However, it could be argued that bulk of the work may be done by the encoder model (here CLIP) as good representations often lead to a good decoder model; see [3], for example.
> > >
> > > In this paper, our language-MILP contrastive learning framework is akin to the CLIP model. In fact, we use a similar contrastive learning framework but in a completely novel way, where we treat MILP and text as different modalities. Our framework shows that indeed we can learn good representations as can be seen by our experiments.
> > >
> > > Given the discussion above, we view the study of the alignment task as an important step towards related tasks, such as directly generating descriptions from MILP instances. While we acknowledge that generating human understandable descriptions from MILP represents the ultimate goal, we, along with the broader community, are not yet at that point. Specifically, previous studies have predominantly focused on either GNNs or LLMs in isolation when studying MILPs. To the best of our knowledge, this work is the first attempt to bridge the gap between recent NLP advancements and state-of-the-art learned MILP models. The disconnect between human understanding and MILP data remains a critical challenge in the optimization community, and this study underscores both the importance and the feasibility of narrowing this gap in the near future. We believe this work provides a meaningful first step toward that objective, similar to how CLIP model's contrastive learning paved the way for text to image generative models such as DALLE by training a decoder model on CLIP embeddings.
> > >
> > > Finally, regarding "However, the experiments focus primarily on aligning existing problems with pre-written descriptions, leaving the generalization ability to unseen MILPs unexplored.", we want to highlight that the MILP instances and the associated descriptions for the MIPLIB dataset and the new MILP-Evolve dataset in [GR1] are both unseen from training. Both experiments demonstrate the generalizability of our method to unseen MILPs.
> > >
> > > *[1] Radford, Alec, et al. "Learning transferable visual models from natural language supervision." International conference on machine learning. PMLR, 2021.*
> > >
> > > *[2] Ramesh, Aditya, et al. "Zero-shot text-to-image generation." International conference on machine learning. PMLR, 2021.*
> > >
> > > *[3] Liu, Haotian, et al. "Visual instruction tuning." Advances in neural information processing systems 36 (2024).*

---

> > > ### Author Response · Authors · 2024-11-27
> > > **Further Response to Reviewer fRvL (2)**
> > >
> > > **Q2: Practical Implications of Integrality Gap Prediction.**
> > >
> > > The LP relaxation is an important aspect in both OR and theoretical CS for designing approximation algorithms for hard optimization problems. For example, integrality gap provides an upper bound on the achievable approximation factors of dual-fitting or primal-dual algorithms, both of which are dominant paradigms in approximation algorithm design literature [1]. Moreover, as mentioned in the paper (Sec. 2.2.1), when the integrality gap is small, one can use the optimal solution to the LP relaxation and round it, say using randomized rounding, to obtain near optimal integral solutions, which is a foundational technique in the design of approximation algorithms [2].
> > >
> > > Regarding “the authors claim this task can reduce solve time or improve solution quality,”  by this sentence we mean the following. The LP relaxation, which is a linear program, is well known to be solvable much more quickly than solving the full MILP instances. There are many fast LP solving algorithms; for example, see [3].  Now, if the integrality gap is small, there could be two ways one can get near optimal or optimal solutions to MILPs: 1) Using the rounding approach we just mentioned. 2) When the integrality gap is small, it also suggests that MILP solvers may converge more quickly to optimal solutions, making it a potential indicator of faster solve times for the corresponding MILP.  However, we do not claim that integrality gap prediction directly improves solution quality, besides conveying the information listed in 1) or 2). We thank the reviewer for asking these clarification questions; we have revised Sec. 2.2.1 of the paper to elaborate along these lines.
> > >
> > > Based on your feedback, we have also revised Sec. 2.2 of the paper to better explain the connection between the tasks. It now states “**Enhancing MILP applicability.** The three learning tasks are complementary and collectively essential for enhancing the applicability of MILP through *understanding, predicting, and accelerating*. In particular, the Language-MILP task helps understand the structure and properties of MILP instances, aiding non-experts in problem comprehension and may further assist experts by deepening their understanding of the problems; the Integrality Gap Prediction task focuses on analyzing solution properties of the MILP instance, potentially allowing instances with tight gaps to be solved via LP relaxation, coupled with rounding algorithms, without fully solving the MILP; the learning to in Branch task enhances MILP solving efficiency through more effective branching, which can have huge time and cost savings in industrial applications”. We hope the reviewer finds the updated version clearer.
> > >
> > > *[1] Williamson, David P., and David B. Shmoys. The design of approximation algorithms. Cambridge university press, 2011.*
> > >
> > > *[2] Raghavan, Prabhakar, and Clark D. Tompson. "Randomized rounding: a technique for provably good algorithms and algorithmic proofs." Combinatorica 7.4 (1987): 365-374.*
> > >
> > > *[3] Cohen, Michael B., Yin Tat Lee, and Zhao Song. "Solving linear programs in the current matrix multiplication time." Journal of the ACM (JACM) 68.1 (2021): 1-39.*

---

> > ### Author Response · Authors · 2024-11-27
> > **Further Response to Reviewer fRvL (3)**
> >
> > **Q3: Contribution Scope and the Role of Foundation Model.**
> >
> > First, as discussed with reviewer dtCu, we are happy to change the title to “*Efficient Multi-Class Learning for Mixed Integer Programming: An LLM-Based Data Augmentation Approach*” in the final version of the paper, if the reviewer thinks this title is more appropriate.
> >
> > Next, we want to reemphasize that our methodology takes a foundation-model like training approach, in a sense that it eliminates the need to train separate models for different MILP problem classes, addressing a significant research gap in the literature. We would like to further provide a concise restatement of our contribution as follows. We hope it can clarify the reviewer’s concern with respect to the contribution of the paper.
> >
> > - **A Foundation Model Approach for Efficient Multi-Class MILP Learning:** We are the first to propose a foundation model training approach for Mixed-Integer Linear Programming (MILP) learning and demonstrate that, a single model, trained on sufficiently diverse MILP problems, can effectively generalize to a variety of unseen MILP classes. Our framework integrates Large Language Models (LLMs) for data generation with Graph Neural Networks (GNNs) for learning MILP instance representations. Unlike prior work that trains GNNs on a limited set of MILP classes, we significantly extend the scope by learning a joint model on a broader and more diverse range of MILP problem classes.
> >
> > - **MILP-Evolve for Data Augmentation:** To address the scarcity of MILP classes, we introduce an LLM-based data augmentation method, MILP-Evolve, that generates diverse MILP problem classes. By combining diverse prompting tailored to the MILP domain, along with parameter search and filtering, MILP-Evolve generates a wide range of MILP classes resembling various real-world optimization scenarios and satisfying targeted properties such as feasibility and nontrivial solve times.
> >
> > - **Comprehensive Framework Evaluation:** We rigorously evaluate our framework across three challenging learning tasks that test different facets of understanding and solving MILP instances. Notably, these tasks involve large-scale MILP instances (e.g., high numbers of variables and constraints) that pose significant challenges even for advanced models like GPT-4o. We demonstrate that our learning method is able to achieve substantial performance improvement across all the learning tasks.
> >
> > - **Broader Impact and Open Science:** Our findings offer new insights into MILP optimization (e.g., diversity, quantity, distribution) and provide a scalable framework to guide future research on effective learning methods for optimization tasks. We further believe that the new language-MILP contrastive learning framework studied in this work can serve as an important stepping-stone for future research on generating natural language descriptions of MILP instances, with potential for broader applicability. We are committed to fully open-sourcing our framework to advance progress in the community.
> >
> > Lastly, we would like further clarification regarding the reviewer’s claim that “MILP-Evolve’s performance is closely tied to seed class selection” and “generalizability to real-world problems remains uncertain”. In particular, we would like to highlight the results on the new MILP-Evolve test set in [GR1], which is a new test set originating from *a completely separate set of seed classes from training*, as well as the results on the MIPLIB dataset, which is *a commonly used benchmark dataset that contains real-world instances also completely unseen from training*. In both cases, our method has significant generalization performance when transferred to these datasets, hence demonstrating the generalizability of our method to real-world problems.
> >
> > **We once again thank the reviewer for their detailed feedback and insights, and look forward to addressing any further concerns they may have.**
> >
> > Best,
> >
> > Authors

---

> > > ### Comment · Reviewer_fRvL · 2024-12-02
> > >
> > > Thank you for the detailed and thoughtful response. I now have a clearer understanding of your work. MILP-Evolve is indeed an interesting contribution, and while it may not fully qualify as a "Foundation Model," it does provide valuable insights and inspiration for future foundation models, particularly in terms of data generation. I believe the paper meets the acceptance threshold for ICLR, and I will raise my score accordingly.

---

> > > > ### Author Response · Authors · 2024-12-02
> > > > **Thank you!**
> > > >
> > > > We sincerely thank the reviewer for their thoughtful review. We are glad that our responses addressed their concerns, and we appreciate the reviewer for raising the score. Thank you once again for your detailed feedback and valuable suggestions throughout the process!

---

### Official Review · Reviewer_zQjg · 2024-11-08

**Soundness:** 3
**Presentation:** 3
**Contribution:** 3
**Rating:** 6
**Confidence:** 3

**Summary:**

This paper considers a novel dataset generation method for learning to solve mixed integer linear programming (MILP), leveraging the large language model (LLM). Given an input MILP instance, this method combines the evolution algorithm and parameter search to compute diversified new instances. The authors consider three tasks: (1) predicting the integrality gap. (2) learning to branch and (3) Aligning MILP problems with natural language to help non-experts.

The authors then tested their method on a dataset called SEED, gathered from the recent popular deep learning for MILP papers. The results showed that their method outperformed all other baselines. Moreover, the attention used on the variables can further improve the transferability.

**Strengths:**

1. Employing LLM to generate diversified MILP instances is novel and helpful for training a foundation model for MILP. The entire MILP space is too huge, so datasets created by humans can only cover a part of it. So, leveraging the power of LLM is a good direction.

2. The authors' commitment to open-source the entire framework is valuable for the entire community.

**Weaknesses:**

1. The dataset test seems to be not that "unseen." You mentioned that you collected MILP problems from eight classes. But you randomly split them after the augmentation. Then, the trained model still learned from all these eight classes. So it would be great if you only use six classes for training, 1 for validation, and 1 for testing. Then this can further show the power of your method.

**Questions:**

1. See the weaknesses (1)

2. One more interesting experiment is to fix the number of training data for your method and the baselines. To be more specific, let N be the number of instances of SEED. Then, we randomly take N / 10 data and use Evolve to generate N instances and call them dataset B. Then, training directly on SEED and this B can further show the power of your model.

---

> ### Author Response · Authors · 2024-11-21
> **Response to Reviewer zQjg**
>
> **We appreciate the reviewer's positive feedback and thank the reviewer for their excellent suggestions on the new experiments to strengthen our work. We have performed the experiments you suggested and have reported the results in general response section [GR1]. We provide  responses to specific questions here.**
>
>
> > The dataset test seems to be not that "unseen." It would be great if you only use six classes for training, 1 for validation, and 1 for testing. Then this can further show the power of your method.
>
> We refer the reviewer to [GR1], where we test the performance on another 50-class test set generated by running MILP-Evolve with six unseen seed classes. Initializing with the Ours pre-trained model yields the best transfer learning performance on this test set. We also note the MIPLIB experiments in the paper also are unseen from training.
>
>
> > One more interesting experiment is to fix the number of training data for your method and the baselines. To be more specific, let N be the number of instances of SEED. Then, we randomly take N / 10 data and use Evolve to generate N instances and call them dataset B. Then, training directly on SEED and this B can further show the power of your model.
>
> Thanks for providing interesting insights and ideas for new ablation studies. First, we would like to provide a clarification for a potential misunderstanding: MILP-Evolve acts on MILP class level instead of MILP instance level; we do not use any seed `instances' (the numerical A, b, c values) but rather the seed class (the code script) to evolve new classes. Each MILP class serves as an instance generator; by setting different randomness of the code script, one can generate unlimited MILP instances from the class. We hope this clarifies our methodology.
>
> Having said that, your comment inspires us to perform the following new ablation study.  For the Integrality Gap Prediction task, we construct different training sets by varying the ratio of seed and instances generated from MILP-Evolve classes. We fix the total number of training and validation instances as 1200 and 300 instances, respectively. We then train a model on each of these training sets and test on the original MILP-Evolve held out test set (main paper Table 1). From the result table below, we see that including more MILP instances from MILP-Evolve (i.e. more diverse MILP classes) improve the performance.
>
> |                          | Seed 100% | Seed 80% + Evolve 20% | Seed 60% + Evolve 40% | Seed 40% + Evolve 60% | Seed 20% + Evolve 80% |
> | ------------------------ | --------- | --------------------- | --------------------- | --------------------- | --------------------- |
> | Deviation ($\downarrow$) | 32.96     | 25.66                 | 23.57                 | 21.67                 | 21.32                 |
> | Correlation ($\uparrow$) | 0.10      | 0.41                  | 0.49                  | 0.55                  | 0.57                  |

---

> ### Author Response · Authors · 2024-12-01
> **Looking Forward to Your Reply**
>
> Dear Reviewer zQjg,
>
> Thank you again for your valuable suggestions. We have conducted detailed experiments and provided thorough discussions in response to your feedback. As the rebuttal period is nearing its conclusion, we wanted to follow up to ensure our responses have adequately addressed the reviewers' concerns. Please let us know if there are any remaining questions or areas where further clarification is needed. We sincerely appreciate the time and effort you have dedicated to reviewing our work. Thank you!
>
> Best,
>
> Authors

---

### Author Response · Authors · 2024-11-21
**General Response to All Reviewers**

We thank all reviewers for their time and effort in reviewing the paper. We are delighted to see that all the reviewers appreciated our work and provided valuable feedback. Here we will answer some common questions that most reviewers asked, and will defer reviewer specific questions to individual responses. Based on your feedback, we provide the following additional experiments strengthen our paper.

- [GR1] We conduct a new set of transfer learning experiments on a dataset consisting of fifty new MILP classes that are not present in the training data.
- [GR2] We provide a detailed explanation on the practical application of Language-MILP Contrastive Learning Task. We conduct additional experiments to compare with GPT-4o to directly interpret the MILP instance files.
- [GR3] We provide an in-depth ablation study on the effect of different seed classes.
- [GR4] Finally, We give more clarifications on why we view our methodology as a foundation model approach.

We further provide detailed response and experiments to each individual reviewer.

---

> ### Author Response · Authors · 2024-11-21
> **[GR1] Generalization abilities of our framework. A New MILP-Evolve Test Set Based on Six Unseen Seed Classes.**
>
> Some of you asked if our model can generalize to MILP classes that have never been seen during training and suggested new experiments. Inspired by Reviewer zQjg's comments, we introduce another test set with 50 classes obtained by running MILP-Evolve on a completely disjoint set of 6 unseen classes*. We ensure that no class in this test set is used in training of our base model and  we will describe how we generate these new MILP classes in the next paragraph. Table (2) summarizes results of our new experiments. It is evident from the new experiments that our pretrained model achieves the best results, highlighting the generalization abilities of our models.
>
> We would also like to use this opportunity to highlight that MIPLIB dataset (already included in the manuscript) is already a strong benchmark for measuring the generalization abilities of our model. This is because, MIPLIB,  a commonly used  MILP benchmark consisting of heterogeneous MILP classes, is totally disjoint from the training dataset. Our results on MIPLIB dataset already shows that our framework generalizes to unseen MILP classes, and our new experiments corroborates on these findings.
>
> Including the new experiments we did during the rebuttal phase, our paper tests generalization abilities of our model on 3 different test datasets that are increasingly more challenging than the previous one: (1) held out classes from the original MILP-Evolve datasets (Table 1 in manuscript), (2) a new set of MILP-Evolve datasets from 6 unseen seed classes (Table 2), and (3) MIPLIB dataset (Table 3).
>
> **Results. All these 3 experiments result in the same consistent behavior: We see that learning on MILP-Evolve data improves the performance across all these levels.**
>
>
> |                    | Integrality Gap |                        | Learning to Branch |                       | Language-MILP         |                        |
> | ------------------ | --------------- | ---------------------- | ------------------ | --------------------- | --------------------- | ---------------------- |
> |                    | Deviation ($\downarrow$)      | Corr. ($\uparrow$) | Acc. ($\uparrow$)    | Top 5 Acc. ($\uparrow$) | 4 Way Acc. ($\uparrow$) | 10 Way Acc. ($\uparrow$) |
> | Train From Scratch | 21.41%          | 0.65                   | 28.93%             | 69.70%                | 72.37%                | 46.50%                 |
> | Seed               | 21.25%          | 0.65                   | 23.11%             | 56.82%                | 72.20%                | 42.45%                 |
> | Seed + Param.      | 25.61%          | 0.52                   | 27.87%             | 68.32%                | 75.17%                | 42.66%                 |
> | Seed + VAE         | 23.40%          | 0.58                   | 25.25%             | 60.81%                | 72.90%                | 44.61%                 |
> | Ours               | **17.98%**          | **0.68**                  | **30.71%**            | **70.33%**               | **77.62%**                | **53.99%**                |
>
> \* **New Experiment Setup.**
>
> - **The New MILP Classes.** We take a different set of 6 unseen seed classes, consisting of Graph Coloring, Job-Shop Scheduling, Protein Folding, Multi-Item Lot Sizing, Bin Packing, and Max Cut. We run MILP-Evolve to slightly expand the new test set to a total of 50 classes. Similar to the MIPLIB experiments, we perform transfer learning of different models to this unseen test set (40% classes for fine-tuning, 60% classes for testing).
> - **Expert Curate / Verified Language Labels for Language-MILP Alignment.** To ensure the quality of language descriptions, we manually verified and modified the linguistic description and made sure the descriptions matches the optimization problem in the testing set.

---

> ### Author Response · Authors · 2024-11-21
> **[GR2] Practical application of Language-MILP Contrastive Learning Task & Comparison with GPT-4o.**
>
> Some of you asked about the significance and practical applications of language-MILP contrastive learning task we study. We answer this question on two fronts: the utility of this task from the perspective of understanding MILPs and the potential of contrastive learning technique itself.
>
> Many open source MILP datasets such as MIPLIB and in many business scenarios, the MILP instance files contain only constraints and variables (the raw $A, b, c$ values in the optimization), which are typically hard to understand and massive in size. Most of these MILP files lack descriptions of the underlying optimization problem and/or not sufficiently meaningful.
> Moreover, we cannot directly feed them into LLMs to interpret and generate language descriptions of the MILP formulations.
>
> To justify this previous claim, as asked by the reviewer (fRvL), we investigated if GPT-4o model can interpret MILPs. We prompt GPT-4o to directly interpret the instance files for the same dataset as in [GR1] (subsample rows up to context length). Unfortunately, out-of-the-box GPT-4o's performance is worse than the model trained with our contrastive loss. The table below summarizes our findings.
>
> |                        | GPT-4o | Train From Scratch | Seed   | Seed + Param. | Seed + VAE | Ours   |
> | ---------------------- | ------ | ------------------ | ------ | ------------- | ---------- | ------ |
> | 4 Way Acc. ($\uparrow$)  | 47.79% | 72.37%             | 72.20% | 75.17%        | 72.90%     | **77.62%** |
> | 10 Way Acc. ($\uparrow$) | 16.81% | 46.50%             | 42.45% | 42.66%        | 44.61%     | **53.99%** |
>
> Hence, this work takes first step with a contrastive learning approach to align GNN embedding of MILP instances with the text embeddings, aiming to provide meaningful interpretations when giving the MILP instances as input. Our results indicate that our approach holds lot of promise.
>
> Given the abstract nature of the MILP instances, we believe any assistance in helping users' understanding of them is crucial. This can help nonexperts to understand the problem and also identify the incorrect formulations. This task also complements our other two tasks which are concerned with solving MILPs rather than understanding. We believe that a foundation model for MILPs that aims to democratize solving MILPs should also have the ability to help users to understand them.
>
> On a more technical side, our language-MILP contrastive learning experiments show that indeed it is possible to align models for this task. We think that this is a novel application of this technique. With more data, we anticipate that our framework can help the community to significantly improve understanding of the MILP instances. Moreover, one can further expand from our work to performance multimodal description generation with the GNN embeddings, which we leave as an interesting (but also challenging) future work.

---

> ### Author Response · Authors · 2024-11-21
> **[GR3] Ablation Study on the Impact of Different Seed Classes.**
>
> Based on Reviewer fRvL's comment, we include ablation studies to analyze the effects of different seed classes.
>
> **Statistics.** The table below shows the proportion of generated classes from each seed class using MIP-Evolve framework.* We see that some seed classes can lead to more generated classes (e.g. Combinatorial Auction (CA)) than others (e.g. Knapsack (KS)). One potential reason could be that our filtering and parameter adjustment procedures can  find good solutions for certain classes more easily than others.
>
> |            | IS    | CA    | KS   | GIS   | NF   | SC   | SAT   | CF   |
> | ---------- | ----- | ----- | ---- | ----- | ---- | ---- | ----- | ---- |
> | Proportion | 10.8% | 27.3% | 4.3% | 22.4% | 7.5% | 6.5% | 11.9% | 9.3% |
>
> \* *Note that for cross-over prompts, we only tracked the trace of the first MILP class, so the number here can be slightly noisy.*
>
> **Impact of different classes on the learning performance.** For each of the eight seed classes, we train separate models on instances sampled using these two methods. (1) All evolved classes based only on a single seed class (**One Seed**), (2) with weighted sampling, where 70% come from all evolved classes based on a single seed class and 30% based on other seed classes (**Weighted**).
> We fix the number of train and validation instances, and compare the test performance on MILP-Evolve held out test set (**Table 1**) and the transfer learning performance to **MIPLIB**.
>
> **Results: We report our findings in the table below; we see that**
>
> - **Table 1, One Seed:** Learning on classes based on one single seed class has limited performance.
> - **Table 1, Weighted:** Classes with a higher proportion in the MILP-Evolve dataset (CA, GIS), when given a higher weight when sampling training set, typically lead to a better test performance on the MILP-Evolve held out set; an exception is Capacitated Facility Location (CF), which has a lower ratio than CA and GIS, but its learned model achieves the best test performance among all models.
> - **MIPLIB, Weighted:** The transfer learning performance when initializing with the different models seem to be similar on the MIPLIB test set, and is worse than Ours performance when trained on instances from all MILP classes.
>
> Finally, these results give more evidence to the primary hypothesis of this paper:  **the importance of having a diverse set of MILP classes from different seed classes to improve the generalization performance.**
>
> |                                       | Table 1, One Seed      |                  | Table 1, Weighted      |                  | MIPLIB, Weighted       |                  |
> | ------------------------------------- | ---------------------- | ---------------- | ---------------------- | ---------------- | ---------------------- | ---------------- |
> |                                       | Deviation ($\downarrow$) | Corr. ($\uparrow$) | Deviation ($\downarrow$) | Corr. ($\uparrow$) | Deviation ($\downarrow$) | Corr. ($\uparrow$) |
> | Seed 0:  Indep. Set (IS)              | 32.66                  | 0.26             | 25.29                  | 0.47             | 25.41                  | 0.47             |
> | Seed 1: Comb. Auction (CA)            | 30.01                  | 0.34             | 21.40                  | 0.53             | 23.44                  | 0.55             |
> | Seed 2: Multiple Knapsack (KS)        | 33.84                  | 0.09             | 24.22                  | 0.49             | 23.60                  | 0.53             |
> | Seed 3: Generalized Indep. Set (GIS)  | 31.74                  | 0.19             | 22.64                  | 0.51             | 25.61                  | 0.49             |
> | Seed 4: Multi-Comm. Network Flow (NF) | 36.49                  | 0.22             | 26.10                  | 0.41             | 24.08                  | 0.52             |
> | Seed 5: Set Cover (SC)                | 34.45                  | 0.11             | 24.80                  | 0.45             | 26.13                  | 0.47             |
> | Seed 6: Max Satisfiability (SAT)      | 43.07                  | 0.20             | 23.52                  | 0.49             | 23.79                  | 0.52             |
> | Seed 7: Cap. Fac. Location (CF)       | 33.00                  | 0.09             | 20.39                  | 0.58             | 25.24                  | 0.51             |
> | Ours (Full Dataset)                   | **20.14**                  | **0.58**             | **20.14**                  |**0.58**             | **21.56%**                 | **0.59**             |

---

> ### Author Response · Authors · 2024-11-21
> **[GR4] Our Contribution**
>
> While creating a framework that can generate diverse and meaningful MILP classes is an important, if not the most important aspect of our work, in our humble opinion, we believe that this study has a substantially broader scope than the LLM-based data generation. *Our key motivation for the work is studying whether a foundation model approach -- that is, pretraining on a large and diverse data that can generalize to a wide variety of downstream tasks -- can be an effective paradigm for MILPs.* We want to highlight that there was no definitive answer to this question prior to our work. As we mentioned in the paper, all the previous work only trained models for specific MILP classes. More importantly, unlike language and image modality, the optimal structure for each MILP instance can be quite different. That is, even if we focus on one MILP problem, say set cover, then the structure of optimal solutions can be quite different. Hence, it is not clear if a DNN trained on a diverse family of MILP classes can generalize.
>
> *Our work shows the feasibility of such an approach and paves the way for more research in this direction.* In our humble opinion, this is a significant contribution of this work and hence we felt justified to call our paper "towards a  foundation model". We acknowledge that our model is not a foundation model (yet), but we are taking steps towards the feasibility of training such a model.
> However, if the reviewers think this title can distract the audience from the main scientific contribution and have better suggestions, we would be happy to reconsider naming the paper differently in the future revisions.
>
> Having said that, gathering training data is an essential step in building a foundation model: for instance, GPT-4o was similarly trained on a large-scale data generation pipeline, resulting in capabilities that have both surprised and greatly benefited its users. Inspired by this philosophy, our work includes a robust process for generating and integrating a diverse set of MILP data to advance research in the field. Beyond data collection, we have developed novel training tasks that allow our model to surpass all baseline models developed from the past years. *Our experiments provide valuable new insights (e.g., diversity, quantity, distribution.) to the optimization community, offering researchers a framework to build more effective models in the future.*
>
> Finally, we believe our proposed methodology and our findings can help the community advance toward production-ready models for MILP that not only accelerate research for optimization experts but also enable non-experts to better understand, plan, and apply optimization techniques effectively. This is also the reason why we are committed to fully open source our framework to advance future research.

---

### Author Response · Authors · 2024-11-21
**General Response: Our Revised Manuscript**

We have revised the paper to incorporate the results and discussions presented in the rebuttal. We colored the updates in blue. Specifically, we have made the following changes:

| Addition                                                                                                                         | Location in the Paper                                                                                            |
| -------------------------------------------------------------------------------------------------------------------------------- | ---------------------------------------------------------------------------------------------------------------- |
| [GR1] Results on a New MILP-Evolve Test Set                                                                                      | Main Paper Sec 5.2, Table 2; Details in Appendix A.3.3                                                           |
| [GR2] Discussion of the Practical Application of the Language-MILP Contrastive Learning Task                                     | Details in Appendix A.3.6; Mentioned in Main Paper Sec. 1.1 and at the end of Sec. 2.2.3                         |
| [GR2] Comparison to GPT-4o                                                                                                       | Details in Appendix A.3.7; Mentioned in Main Paper Sec. 1.1 and 5.1.                                             |
| [GR3] Ablation Study on the Impact of Different Seed Classes                                                                     | Appendix A.3.4                                                                                                     |
| [GR4] Discussion of Our Contribution Towards Foundation Models for MILPs                                                         | Sec. 1: we update the introduction to put more emphasis on our contribution towards foundation models for MILPs. |
| Reviewer zQjg: an Ablation Study on the Effect of Mixing Different Fractions of Seed and MILP-Evolve Generated Instances         | Appendix A.3.5                                                                                                     |
| Reviewer fRvL: Discussion of the Connections between the Learning Tasks                                                           | Main Paper, at the end of Sec. 2.2.3                                                                             |
| Example Language Descriptions for the Language-MILP Alignment Tasks; An Example of the Prompt and Answer with GPT-4o (for [GR2]) | Appendix A.3.7 (page 36 - 38)                                                                                     |

*Due to space constraint, we moved the instance statistics visualization (previously, Fig. 6) from the main paper to Appendix A.3.1 (currently, Fig. 11), and we updated the reference in the main paper accordingly. If the reviewers have strong preferences regarding this change, We are happy to consider alternatives and also welcome suggestions from the reviewers.*

---

### Meta-Review · Area_Chair_o9t5 · 2024-12-21

**Metareview:**

This paper presents a framework leveraging LLMs for generating diverse MILP instances to improve multi-class learning in optimization. The work demonstrates solid performance improvements across three tasks with extensive experimental validation. While all reviewers acknowledged the paper's clear presentation and practical utility, there were initial concerns about methodological novelty, title appropriateness, and practical significance of the language alignment task. Through comprehensive author responses, including new experiments and detailed clarifications, most concerns were addressed, though some reservations about technical innovation remained. The final reviewer consensus suggests the paper meets the acceptance threshold.

**Additional Comments On Reviewer Discussion:**

Initially, reviewers raised concerns about the paper's scope, novelty, and practical applications. The authors provided extensive responses. This led to constructive dialogue, particularly with one reviewer who initially had significant concerns but was convinced by the authors' thorough responses. While some reviewers maintained reservations about technical novelty, they acknowledged the paper's value to the optimization community, leading to a consensus that the work, while perhaps not groundbreaking in methodology, represents a valuable contribution worthy of publication.

---

### Decision · Program_Chairs · 2025-01-22

Accept (Poster)